# OpenAGI: When LLM Meets Domain Experts

**Yingqiang Ge**
Rutgers University

**Wenyue Hua**
Rutgers University

**Kai Mei**
Rutgers University

**Jianchao Ji**
Rutgers University

**Juntao Tan**
Rutgers University

**Shuyuan Xu**
Rutgers University

**Zelong Li**
Rutgers University

**Yongfeng Zhang**[*]
Rutgers University

## Abstract

Human Intelligence (HI) excels at combining basic skills to solve complex tasks. This capability is vital for Artificial Intelligence (AI) and should be embedded in comprehensive AI Agents, enabling them to harness expert models for complex task-solving towards Artificial General Intelligence (AGI). Large Language Models (LLMs) show promising learning and reasoning abilities, and can effectively use external models, tools, plugins, or APIs to tackle complex problems. In this work, we introduce **OpenAGI**, an open-source AGI research and development platform designed for solving multi-step, real-world tasks. Specifically, OpenAGI uses a dual strategy, integrating standard *benchmark tasks* for benchmarking and evaluation, and *open-ended tasks* including more expandable models, tools, plugins, or APIs for creative problem-solving. Tasks are presented as natural language queries to the LLM, which then selects and executes appropriate models. We also propose a Reinforcement Learning from Task Feedback (RLTF) mechanism that uses task results to improve the LLM's task-solving ability, which creates a self-improving AI feedback loop. While we acknowledge that AGI is a broad and multifaceted research challenge with no singularly defined solution path, the integration of LLMs with domain-specific expert models, inspired by mirroring the blend of general and specialized intelligence in humans, offers a promising approach towards AGI. We are open-sourcing the OpenAGI project's code, dataset, benchmarks, evaluation methods, and the UI demo to foster community involvement in AGI advancement: `https://github.com/agiresearch/OpenAGI`.

## 1 Introduction

The acquisition and reuse of skills is a fundamental aspect of human intelligence that enables the formation of complex skills to address novel or intricate problems [19, 4, 57]. We posit that machine intelligence should incorporate this capacity to synthesize various skills by composing them into complex skills for complex task-solving. In computer science parlance, each skill is referred to as a domain expert "model" – a reusable tool, module, network, plugin, or API with a defined function. The domain expert models can be synthesized into a larger "plan" for performing more complex tasks. The model synthesis process is adaptable to the input or task, such that for a given task, the models are synthesized into the most suitable plan to address the task at hand. As a result, different inputs or tasks may necessitate distinct synthesized models as a plan for task-solving.

Recent advances in Large Language Models (LLMs) have showcased exceptional learning and reasoning capabilities, rendering them well-suited for selecting, synthesizing, and executing external expert models to address complex tasks. These LLMs, such as GPT series [32, 2], LLaMA series [45, 44] and T5 series [33, 8], have exhibited a profound understanding of natural language and the

---

[*]{yingqiang.ge,wenyue.hua,kai.mei,jianchao.ji,juntao.tan,shuyuan.xu,zelong.li,yongfeng.zhang}@rutgers.edu

37th Conference on Neural Information Processing Systems (NeurIPS 2023) Track on Datasets and Benchmarks.

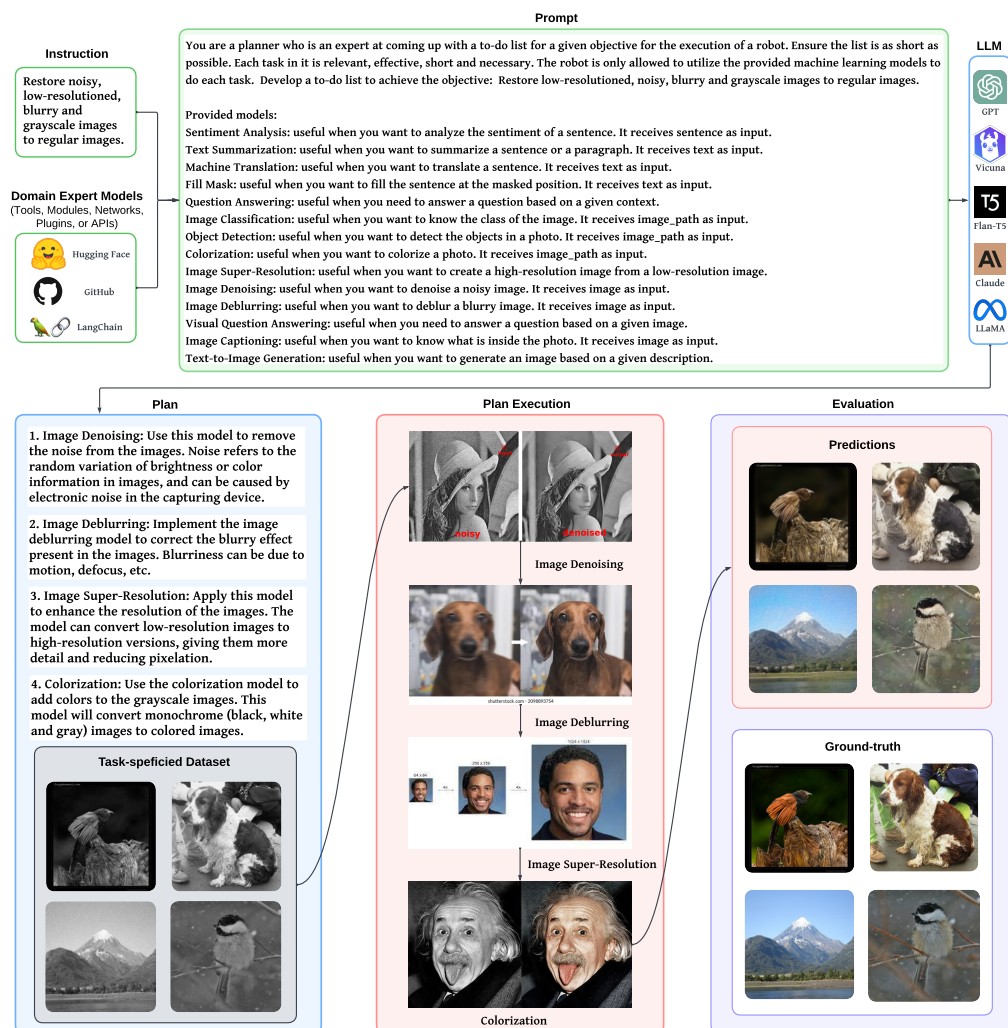

Figure 1: An example of benchmark tasks, which shows the OpenAGI pipeline. OpenAGI generates a task-solving plan for the input task described in natural language (using GPT-3.5 in this example, but can be other LLMs such as GPT-4, Vicuna, Flan-T5, Claude-2, and LLaMA-2), executes the plan with domain expert models, tools, APIs, and then conducts evaluation for the plan execution results.

ability to generate coherent and contextually relevant responses. This has opened up new possibilities for their application in complex tasks involving multi-modality data, such as image and text processing, as well as the integration of domain-specific knowledge. In this process, LLMs play a crucial role as they can understand and generate natural language, which helps AI to better comprehend and handle various problems. By integrating knowledge and skills from different domains, **Open-domain Model Synthesis (OMS)** holds the potential to drive the development of artificial general intelligence (AGI), enabling AI to solve a diverse array of problems and tasks. Despite acknowledging the complexity and lack of a defined path towards AGI, the combination of LLMs and domain-specific expert models, inspired by the interplay of general and specialized intelligence in humans, provides a promising direction [19]. However, the current research field, despite initial attempts, presents several significant challenges: 1) **Extensibility**: Several existing works employ a fixed number of models, such as WebGPT [25] and ToolFormer [40], resulting in difficulties when attempting to expand their capabilities; 2) **Nonlinear Task Planning**: The majority of current research is limited to solving tasks with linear task planning solutions [49, 18], meaning that each sub-task must be completed before the next sub-task can start. However, linear planning of models may not suffice for solving complicated tasks, besides, many tasks involve multiple multi-modal inputs; 3) **Quantitative Evaluation**: Many existing works only provide qualitative results, such as HuggingGPT [42]. This makes it difficult to assess the planning capabilities of LLMs to determine whether the strategies employed are optimal.

In order to mitigate the above limitations, we develop a platform that encompasses a diverse array of domain-specific expert models and intricate multi-step tasks with single or multiple multi-modal inputs. Furthermore, to promote the community's long-term advancement and assessment of AGI's

abilities, we open-source all code and datasets, and hence, name this platform **OpenAGI**. A toy example, showing the entire pipeline of OpenAGI, is depicted in Fig. 1. Specifically, 1) a natural language instruction of a specific task is given; 2) the instruction is augmented by manually designed prompt and then fed as input into LLM to generate a plan; 3) the expert models are selected and synthesized based on the generated plan, and subsequently executed to process the data samples; 4) the task-solving ability of the LLM can be evaluated by comparison between the output and the ground-truth labels or through human evaluation.

OpenAGI embodies a dual approach to address diverse requirements–**benchmark tasks** and **open-ended tasks**. On the one hand, we have incorporated benchmark tasks, each supported by task-specific datasets and evaluation metrics. This inclusion provides researchers with a consistent platform to assess and compare the performance of various models, stimulating continuous improvement and competitive innovation. For benchmark tasks, as depicted in Fig. 1, we utilize a selection of expert models derived from esteemed libraries such as Hugging Face's transformers and diffusers, as well as from GitHub repositories, thereby easily facilitating the expansion of our model set. Additionally, the datasets have been meticulously selected to align with or resemble the training datasets of the respective models. We then implement a variety of data augmentation techniques to enhance these original datasets, enabling the construction of sophisticated multi-step tasks designed to assess the planning and task-solving capabilities of a given LLM. On the other hand, OpenAGI also offers open-ended tasks that utilize a variety of expandable models. These tasks open the door to creativity and imaginative problem-solving, enabling the exploration of innovative solutions that may not emerge within more constrained task frameworks. For open-ended tasks, as depicted in Fig. 2, which is designed to accommodate a broader spectrum of needs, we further include LangChain to provide additional expert models, such as Google Search, Wikipedia, Wolfram Alpha and so on. Indeed, relying solely on input text for learning proves insufficient for LLMs when faced with real-world tasks. In order to improve its performance, we introduce a mechanism referred to as **Reinforcement Learning from Task Feedback (RLTF)**. This approach capitalizes on the performance feedback procured from tasks following the execution of the solution devised by the LLM. Consequently, the RLTF mechanism effectively refines the LLM's planning strategy, resulting in an enhanced and more adaptive system. In summary, the key contributions of the work include:

- We introduce OpenAGI, an AGI research platform, specifically designed to offer complex, multi-step tasks accompanied by their respective datasets, evaluation methods, and a diverse range of extensible models which can be synthesized to effectively solve these tasks. The purpose of this platform is to aid in the quantification of the overarching planning and task-solving abilities of LLMs. OpenAGI embraces AGI by focusing on LLM-driven, (open-domain) model synthesis, predominantly utilizing models and datasets on Hugging Face, GitHub and LangChain.

- We propose the LLM+RLTF approach for OpenAGI, which leverages a Large Language Model as a controller to select, synthesize and execute various external expert models for complex task-solving. The feedback obtained from these tasks is then employed to refine the LLM's planning strategy, thereby enhancing the LLM's overall performance and task-solving ability.

- We evaluate both open-source and closed-source LLMs with differing scales under distinct learning schema and the OpenAGI pipeline. Our findings suggest that even smaller-scale LLMs, when paired with an appropriate learning schema such as RLTF, are able to possess the potential to outperform competitors that equip a significantly greater magnitude of model parameters.

## 2   Related Work

### 2.1   Large Language Model and AI Agents

With the advancement of highly parallelizable transformer architectures, pre-trained language models (PLMs) have demonstrated remarkable capabilities in comprehending, generating, and manipulating natural language [31, 24]. These models are pre-trained on a large corpora of text data and commonly fine-tuned for specific downstream tasks. Shortly, the scaled-up PLMs, known as Large Language Models (LLMs) [34, 2, 27, 6, 55, 45], encompassed a substantially greater number of parameters and leveraged vast amounts of training data. Consequently, LLMs exhibited an enhanced capacity to learn intricate language patterns and structures, along with a notable reasoning ability, leading to superior performance across diverse natural language processing tasks [2, 45, 55, 6, 5, 30, 14, 52]. Apart from the above superiority, LLMs may occasionally produce seemingly plausible yet inaccurate predictions and face challenges when addressing problems that require specialized domain

expertise [23]. Consequently, the emerging field of Augmented Language Models (ALMs) focuses on addressing the limitations of conventional LLMs [8, 6, 2] by equipping them with enhanced reasoning capabilities and the ability to employ external resources [23]. The process of reasoning involves breaking down intricate assignments into smaller, more manageable sub-tasks that can be independently or collaboratively tackled by LLMs with the assistance of tools. What's more, LLMs can also invoke external tools or models to accomplish the relevant tasks. For example, ToolFormer [40] introduces external API tags within text sequences, facilitating LLMs' access to external tools. Visual ChatGPT [51] combines ChatGPT with Visual Foundation Models (VFMs) such as Transformers, ControlNet, and Stable Diffusion, which acts as a bridge between users, allowing them to communicate via chat and generate visuals. HuggingGPT [42] integrates the Hugging Face hub with task-specific models around ChatGPT to tackle AI tasks. ChatGPT for Robotics [47] employs ChatGPT for a wide array of robotics tasks through strategic prompt engineering. Besides, several open-sourced GitHub repositories are related to this topic, such as BabyAGI and AutoGPT. Notably, AutoGPT [15] is an automated agent, which is designed to set multiple objectives, break them down into relevant tasks, and iterate on these tasks until the objectives are achieved. Augmented language models may use these enhancements separately or joint them in a specific order to finish the specific task, which ultimately results in superior generalization capabilities.

Different from other works, we propose OpenAGI, an open-source AGI research and development platform designed to address the challenges commonly encountered in existing works, such as extensibility, nonlinear task planning, and quantitative evaluation. Furthermore, we introduce innovative methods into the learning schema of LLMs, including Reinforcement Learning from Task Feedback (RLTF) and nonlinear task planning, which aims to address challenges on out-of-distribution (OOD) generalization, optimal task planning, and AI's self-improvement (please see Sec. A.1 in supplementary materials for an extended discussion on these problems). We hope the OpenAGI platform can facilitate the open and long-term development and evaluation of AGI abilities in the community.

## 2.2 Reinforcement Learning from Human Feedback (RLHF)

To better align Large Language Models (LLMs) with human values, Reinforcement Learning from Human Feedback (RLHF) has been introduced [7, 58], which fine-tunes LLMs by collected human feedback, effectively enhancing alignment criteria such as helpfulness, honesty, and harmlessness. At its core, RLHF deploys reinforcement learning (RL) algorithms, notably Proximal Policy Optimization (PPO) [41], to tailor LLMs to this feedback via a reward model. Importantly, this approach actively involves human oversight in the training process, exemplified by notable models such as InstructGPT [27]. Nonetheless, the efficacy of RLHF is contingent upon the quality of feedback from adept labelers, rendering its practical implementation challenging [13, 29]. Consequently, there is an imperative to refine the RLHF framework to diminish the dependency on manual labeling and explore innovative, efficient annotation methodologies that ensure data integrity. Compared with RLHF, the proposed RLTF gets task feedback to supply information that guides the learning direction of LLMs, resulting in improved and more efficient strategies, which does not require human intervention.

## 3   The OpenAGI Platform

OpenAGI includes a wide range of features tailored to various needs. One key component is its benchmark tasks, detailed in Sec. 3.1, a particularly valuable tool for researchers. These tasks come equipped with task-specific datasets and evaluation metrics. This makes it possible for researchers to evaluate the performance of different LLMs in a structured and uniform manner, offering insights into their efficacy and potential areas for improvement. In addition to benchmark tasks, OpenAGI also offers open-ended tasks, detailed in Sec. 3.2. These tasks allow for a greater degree of creativity and imagination, breaking away from conventional constraints to enable more exploratory research. We believe this combination of structured benchmark tasks and flexible open-ended tasks makes OpenAGI a robust and versatile platform that can cater to a diverse array of research requirements.

## 3.1   Benchmark Tasks

For benchmark tasks, our goal is to provide the community a valuable tool to evaluate the planning abilities of LLMs for complex, multi-step tasks. Specifically, instead of building complicated, multi-step tasks from scratch, we first explore the domain expert models (Sec. 3.1.1) that can be used as building blocks, then introduce how we create such tasks based on them (Sec. 3.1.2).

### 3.1.1 Domain Expert Model Set

We now present the domain tasks and the corresponding models that can be employed in our platform. This set is designed to be flexible, allowing users to easily incorporate their own domain tasks and models. Our domain tasks are as follows:

- **Language-related Models**: **Sentiment Analysis** classifies the sentiment polarity of a given sentence [1]; **Text Summarization** creates a text summary that represents the most important or relevant information within the original text content [20]; **Machine Translation** converts a sentence from a source language to a target language [34]; **Fill Mask** involves replacing masked words within a given text [22]; **Question Answering (QA)** provides a textual answer of a question based on the given context [39].

- **Vision-related Models**: **Image Classification** aims to comprehend an entire image as a whole and assign it to a specific label [12]; **Object Detection** identifies and localizes specific objects within an image by detecting their instances of a particular class [3]; **Colorization** refers to the technique of adding plausible color information to monochromatic photographs or videos [54]; **Image Super-resolution** generates a high-resolution (HR) image from a low-resolution (LR) image [10]; **Image Denoising** aims to remove unwanted noise from an image while preserving its important features [53]; **Image Deblurring** aims to recover a clear image from a blurred input image [53].

- **Vision-Language Models**: **Visual Question Answering (VQA)** involves answering questions based on an image [48]; **Image Captioning** generates textual descriptions of the visual content depicted in an image; **Text-to-Image Generation** generates images from a given input sentence or sequence of words [36].

The details of the corresponding models are shown in Tab. A.1, A.2 and A.3 in supplementary materials. After selecting the domain expert models, choosing the raw datasets becomes a more straightforward process, provided that we need to ensure proper alignment between the datasets and the domain expert models' training sets. Raw datasets are provided as follows: **ImageNet-1K** [38], **Common Objects in Context (COCO)** [21], **CNN/Daily Mail** [26], **Stanford Sentiment Treebank (SST2)** [28], **TextVQA** [43], **Stanford Question Answering Dataset (SQuAD)** [35]. More details about theses datasets can be found in Sec. A.2 in supplementary materials.

### 3.1.2 Multi-step Tasks and Corresponding Datasets Construction

A multi-step task, as the name suggests, refers to a complex problem that cannot be solved in one simple step. It necessitates several sub-processes or stages, each requiring a particular type of problem-solving skill, in other words, domain expert model. In order to construct such complex, multi-step tasks, we introduce several commonly-used data augmentation methods, which are **Gaussian Blur**, **Gaussian Noise**, **Grayscale**, **Low Resolution**, **Translation**, **Word Mask**, to augment the raw dataset. More details about these methods can be found in Sec. A.3 in supplementary materials.

For the purpose of our study, we have sorted these tasks into six primary categories according to the modalities of their inputs and outputs:

- *Image in, image out*: In these tasks, images undergo several transformation stages. An example could be a task that involves "Denoising and enhancing the resolution of a low-resolution, noisy image". Here, the multi-step process entails image denoising followed by super-resolution.

- *Image in, text out*: These tasks usually involve interpreting the content of images. For example, "Detect objects in an image and describe them in a sentence" requires object detection followed by text generation.

- *Text in, image out*: Tasks under this category may include generating an image based on textual descriptions, such as "Create a graphical representation of the room described in the given text", demanding text understanding and image generation steps.

- *Text in, text out*: These tasks engage in text transformation or interpretation. For instance, "Translate a paragraph from English to German and summarize it in English" requires two steps - translation and summarization.

- *Image-text pair in, text out*: These tasks deal with complex interplay between visual and textual data. For example, "Given an image and a question about the image in English, answer the question in German." This task includes image-text understanding, question answering, and translation.

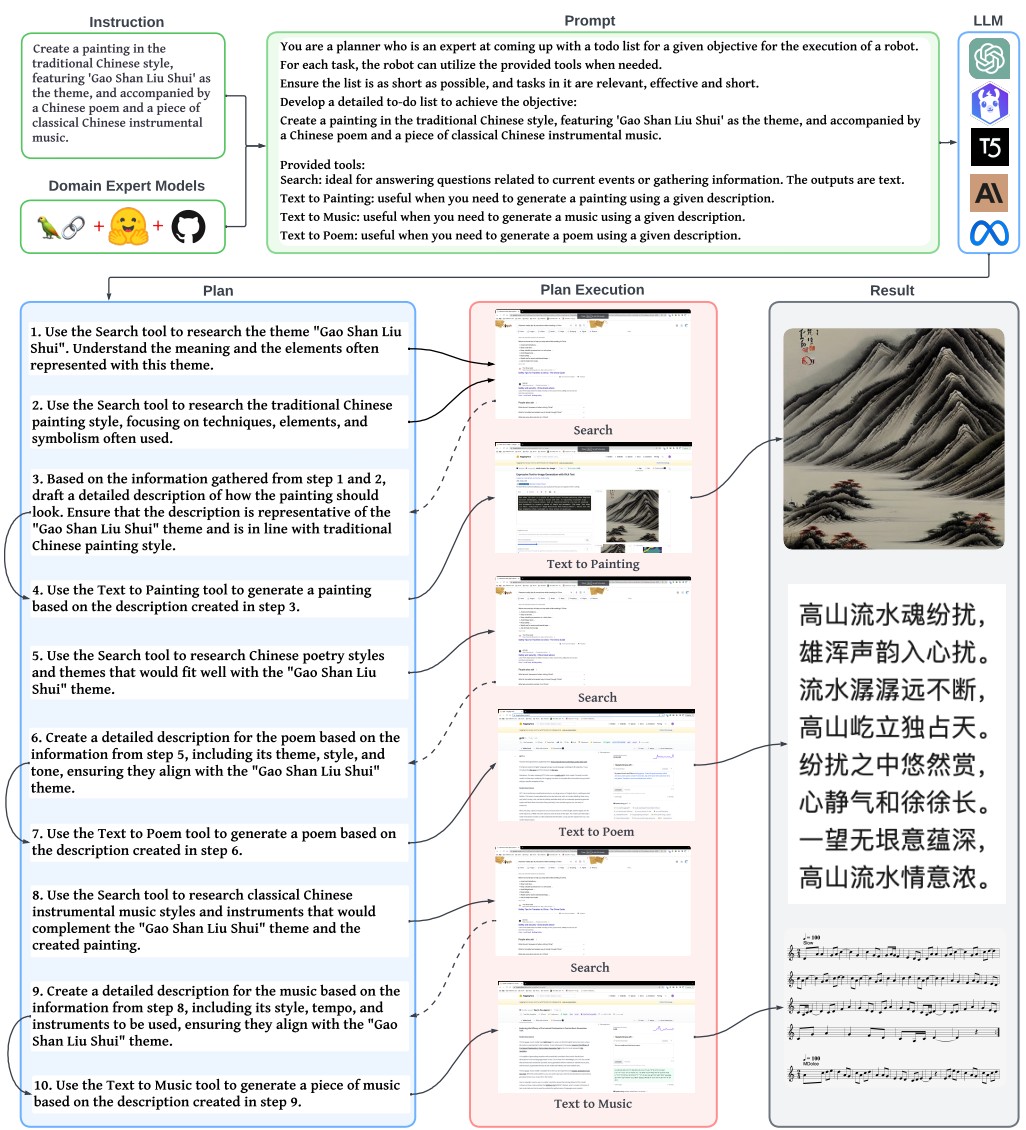

Figure 2: An example of open-ended tasks, which instructs OpenAGI to create an artwork given the theme "Gao Shan Liu Shui" (translating to "High Mountain and Flowing Water" in English). OpenAGI generates a non-linear (tree-structured) plan for the task with GPT-3.5, and then executes the plan with expert models to create a painting, a poem, and a piece of music for the theme.

- *Text-text pair in, text out*: These tasks can involve comparison, synthesis, or information extraction from two text inputs. For instance, "Given two reviews of a movie in English, translate them into German and provide a summary."

In total, we have devised 185 multi-step tasks, of which 117 tasks maintain a linear task structure with steps following a simple sequence, while the remaining 68 tasks exhibit a non-linear task structure, where steps might be performed concurrently or in a complex order. Among these categories, tasks such as Question Answering (QA) and Visual Question Answering (VQA), involving multiple or even multi-modal inputs, are notably complex and defy simple, linear task planning solutions. For a comprehensive view, we provide example tasks and their input and output data samples in Tab. A.4 of the supplementary materials. Additionally, a complete list of the task descriptions, accompanied by their estimated difficulty levels, can be found in Tab. A.5 within the supplementary materials.

### 3.1.3 Evaluation Metrics

Given that the benchmark tasks of OpenAGI comprise a diverse range of domain tasks with multi-modal data, we classify them according to domain tasks as well as input and output types. We then

assess their performance using the following three metrics based on their categories: **CLIP Score** [16], **BERT Score** [56] and **ViT Score**[2] (more details can be found in supplementary). In particular, we employ the CLIP Score only for Text-to-Image Generation-based tasks, the BERT Score is utilized to assess tasks with text outputs, and the ViT score is applied to measure image similarity for the remaining tasks with image outputs. We also normalize the BERT and CLIP scores.

## 3.2 Open-ended Tasks

Open-ended tasks necessitate an elevated degree of creative and imaginative capacity, as they deviate from conventional constraints to stimulate more exploratory research. These tasks are designed to accommodate a broad spectrum of needs, as illustrated in Fig. 2. To achieve this, LangChain is integrated to provide additional expert models from renowned sources such as Google Search, Wikipedia, Wolfram Alpha, and more. Crucially, these models offer extendability, ensuring that open-ended tasks are not confined to specific guidelines or performance metrics. To exemplify this process, Fig. 2 elucidates how OpenAGI is directed to create a traditional Chinese painting with "Gao Shan Liu Shui" (translating to "High Mountain and Flowing Water" in English) as its theme. The process is enriched with the addition of a generated ancient Chinese poem and a piece of music that harmonize with the painting. To effectively deliver on this instruction, OpenAGI first conducts an online search to comprehend the historical narrative of "Gao Shan Liu Shui". Sequentially, the painting, poem, and music are generated in a step-by-step fashion, leveraging the collaboration between expansive language models and domain-specific expert models. The final product – a coherent artistic ensemble of painting, poem, and music – successfully resonates with the underlying ancient narrative, demonstrating the efficacy of this approach in open-ended tasks. More examples are provided in supplementary.

## 4 Reinforcement Learning from Task Feedback (RLTF)

While learning solely from input text is a powerful method for training LLMs, it is not sufficient for handling real-world tasks that require a deeper understanding of context and environment. One potential method to improve the capabilities of LLMs is to incorporate reinforcement learning (RL) techniques. By leveraging the strengths of RL, LLMs can gain additional insights from trial-and-error experiences. This leads to more robust and adaptive models, especially in situations where labeled data is scarce or when tasks involve physical interactions. In this work, we propose Reinforcement Learning from Task Feedback (RLTF), shown in Fig. 3, which utilizes task feedback to supply more information that guides the learning direction of LLMs, resulting in improved and more efficient strategies. We choose to use REINFORCE [50] in this work and more details about the algorithm are provided in Sec. A.6 in supplementary.

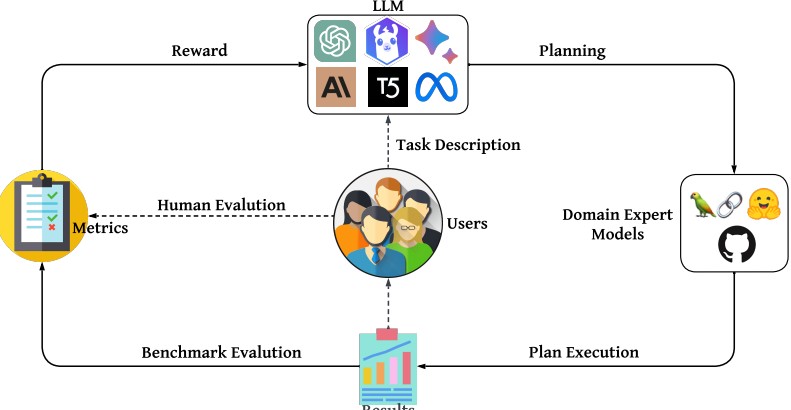

Figure 3: An illustration of the RLTF mechanism.

## 5 Experiments

### 5.1 Backbone LLMs

- **GPT-3.5-turbo.** The GPT (Generative Pre-trained Transformer) series [2] consists of advanced language models. In this work, we use the GPT-3.5-turbo-0301 snapshot.

---

[2]https://colab.research.google.com/github/huggingface/notebooks/blob/main/examples/image_similarity.ipynb

- **Claude-2.** Claude-2 [9] is an transformer LLM trained with unsupervised learning and RLHF.
- **GPT-4.** GPT-4 is a follow-up version of GPT-3.5, which is more powerful than its predecessors. In this work, we use the GPT-4-0613 snapshot.
- **Flan-T5-Large.** Flan-T5 [8] is a series of language models which are fine-tuned using a technique called instruction fine-tuning. Flan-T5-Large has 770 million parameters.
- **Vicuna-7B.** Vicuna [5] is an open-source chatbot trained by fine-tuning the LLaMA [45] model with user-shared conversations. In this work, we use the 7-billion size model of Vicuna.
- **LLaMA-2.** LLaMA-2 [46] is a successor to the original LLaMA model, it is significantly more powerful. In this work, we use the 13-billion size model.

Overall, we include three closed-source LLMs (GPT-3.5-turbo, Claude-2 and GPT-4) as well as three open-source LLMs (Flan-T5-Large, Vicuna-7B and LLaMA-2-13B).

## 5.2 Learning Schema of LLMs

We employ the following LLM learning schema for experimentation.

- **Zero-shot Learning (Zero)** directly inputs the prompt to the LLM.
- **Few-shot Learning (Few)** presents a set of high-quality demonstrations, each consisting of both input and desired output, on the target task. As the model first sees good examples, it can better understand human intention and criteria for what kinds of answers are wanted.
- **Fine-tuning** involves using manually labeled data samples as additional training signals to refine and adapt pre-trained LLMs to specific tasks or domains.
- **RLTF** is our proposed method in Sec. 4, which further utilizes the RL method to tune the fine-tuned LLMs with human labelled data.

We employ Low-Rank Adaptation (LoRA) [17] to optimize all open-source LLMs across both the Fine-tuning and RLTF learning schema.

## 5.3 Planning Solution Parser

To transform the original LLM output into a viable task planning solution, we use a parser built on GPT-3.5. The prompt we employed reads as follows: "You are a key phrase extractor who is able to extract potential module names from the given context. You have already known all the module names in the full module list. The full module list is: [Image Classification, Colorization, Object Detection, Image Deblurring, Image Denoising, Image Super Resolution, Image Captioning, Text to Image Generation, Visual Question Answering, Sentiment Analysis, Question Answering, Text Summarization, Machine Translation]. Given the following context: '{}'. Please extract a module sequence from this context and remove module names which do not exist in the full module list from this sequence. Output the module sequence after filtering as the format of 'module: module1, module: module2, module: module3, etc...'." Once this prompt is executed on the LLM's original text output, a task planning solution will be generated which consists of a multi-step solution of the problem.

## 5.4 Datasets

Considering the fact that the imbalanced number of tasks with different input and output modalities could lead to skewed measurement results, we select the tasks in OpenAGI to compose the training set. In particular, we randomly select 10% of tasks, along with their corresponding datasets, based on input and output modalities for training purposes. For few-shot, fine-tuning and RLTF, we supply manually curated, feasible solutions as ground-truth labels. In the case of RLTF, we employ the fine-tuning checkpoint as a reasonable initialization for LLMs and use constrained beam search [11, 37] to reduce the likelihood of producing infeasible solutions (details can be found in Sec. A.7 in supplementary). Moreover, we choose an additional 10% of tasks, adhering to the same selection criteria as mentioned above, to serve as the test set.

## 5.5 Experimental Analysis

The main experimental results are tabulated in Tab. 1 and 2, showcasing the results for closed-source and open-source LLMs, respectively. The overall performance is calculated as the average of CLIP,

BERT and ViT scores. Here, only the task descriptions of the benchmark tasks are fed into LLMs (additional information, such as the input prompt and LLMs' outputs, is provided in Fig. A.4 and A.5 in supplementary). Broadly speaking, closed-source LLMs demonstrate superior performance on OpenAGI tasks, with GPT-4 leading the pack under both zero- and few-shot scenarios. In the open-source category, LLaMA-2-13B takes the lead, consistently posting top results across various learning schema—the performance possibly influenced by its larger model size. Notably, open-source LLMs significantly benefit from the tuning methods, particularly Fine-tuning and RLTF. These methods mark noticeable enhancements for Flan-T5-Large, Vicuna-7B, and LLaMA-2-13B when compared with zero-shot and few-shot learning schema. In fact, each of these open-source models hits its pinnacle under the RLTF approach. Conclusively, with RLTF tuning, the performance of LLaMA-2-13B approaches that of GPT-3.5, illustrating its potential.

Table 1: OpenAGI task-solving performances under different settings for three closed-source LLMs. Boldface denotes the highest score under each learning schema.

| Metrics | GPT-3.5-turbo | | Claude-2 | | GPT-4 | |
|---|---|---|---|---|---|---|
| | Zero | Few | Zero | Few | Zero | Few |
| CLIP Score | 0.0 | 0.0 | 0.0 | 0.2543 | 0.0 | 0.3055 |
| BERT Score | 0.1914 | 0.3820 | 0.2111 | 0.5038 | 0.2076 | 0.6307 |
| ViT Score | 0.2437 | 0.7497 | 0.4082 | 0.5416 | 0.5058 | 0.6480 |
| Overall | 0.1450 | 0.3772 | 0.2064 | 0.4332 | **0.2378** | **0.5281** |

Table 2: OpenAGI task-solving performances under different settings for three open-source LLMs. Boldface denotes the highest score under each learning schema.

| Metrics | Flan-T5-Large | | | | Vicuna-7B | | | | LLaMA-2-13B | | | |
|---|---|---|---|---|---|---|---|---|---|---|---|---|
| | Zero | Few | Fine-tuning | RLTF | Zero | Few | Fine-tuning | RLTF | Zero | Few | Fine-tuning | RLTF |
| CLIP Score | 0.0 | 0.0 | 0.0 | 0.0 | 0.0 | 0.0 | 0.0 | 0.0 | 0.0 | 0.0612 | 0.0608 | 0.1220 |
| BERT Score | 0.0 | 0.2488 | 0.0 | 0.0655 | 0.0513 | 0.0 | 0.1212 | 0.1756 | 0.0986 | 0.2281 | 0.1570 | 0.2401 |
| ViT Score | 0.0 | 0.0 | 0.6316 | 0.6978 | 0.1704 | 0.4285 | 0.5507 | 0.7300 | 0.3614 | 0.2558 | 0.6723 | 0.7584 |
| Overall | 0.0 | 0.0829 | 0.2105 | 0.2544 | 0.0739 | 0.1428 | 0.2239 | 0.3018 | **0.1533** | **0.1817** | **0.2967** | **0.3735** |

## 5.6 Effect of Prompts

We design two types of prompts combined with different levels of model description to test LLMs' zero-shot performances. The first, Prompt-1, only combines the task description with the model names, while the second, Prompt-2, integrates the task description with comprehensive model descriptions, detailing model usage, input, and output types (additional information about these two prompts is provided in Fig. A.6 in supplementary). We analyze the results in Tab. 3 and 4 in conjunction with the previous zero-shot results in Tab. 1 and 2. Compared to the original prompt that only uses task description to generate the results in Tab. 1 and 2, it is evident that in most cases, the closed-source LLMs, such as GPT series and Claude-2, tend to outperform when provided with detailed model-related information as seen in Prompt-1 and Prompt-2. In contrast, open-source LLMs, whose understanding and reasoning capacity may be weaker than those huge closed-source models, appear to be misled by the ambiguous details in Prompt-1 and Prompt-2 during the model selection process. Overall, detailed prompts can assist in improving the zero-shot performance to a certain degree, depending on the specific model. However, they may not be as potent as other training scenarios for smaller size LLMs, such as fine-tuning or RLTF.

## 5.7 Case Study of Non-linear Planning

We qualitatively evaluate LLMs' capability of non-linear task planning. Fig. 4 illustrates the responses of GPT-3.5, Vicuna-7B and Flan-T5-Large to Prompt-2. The given task description requires the LLM to answer a query posed in English about a given noisy, blurry, and gray-scale image in German. It can be observed from the results that the performance of the models varies significantly. Flan-T5-Large, for instance, demonstrates a struggling comprehension of the query, while Vicuna-7B's answer incorporates all the provided models in an attempt to resolve the task. GPT-3.5 successfully comprehends the task and consequently delivers a reasonable plan. The plan generated by this model is notably non-linear, and it instructs to employ a Visual Question Answering (VQA) model with the English query and processed image as inputs in steps 1 and 2 in order to accomplish the task. Similarly, another task scenario is demonstrated in Fig. 2, which is an open-ended task with

Table 3: Zero-shot task-solving performances under various prompts for three closed-source LLMs.

| Metrics | GPT-3.5-turbo | | Claude-2 | | GPT-4 | |
|---|---|---|---|---|---|---|
| | Prompt-1 | Prompt-2 | Prompt-1 | Prompt-2 | Prompt-1 | Prompt-2 |
| CLIP Score | 0.0 | 0.0 | 0.0 | 0.0 | 0.0 | 0.0 |
| BERT Score | 0.2106 | 0.3013 | 0.4088 | 0.2333 | 0.4402 | 0.5595 |
| ViT Score | 0.0 | 0.2710 | 0.6816 | 0.7957 | 0.5497 | 0.5565 |
| Overall | 0.0702 | 0.1907 | 0.3635 | 0.3430 | 0.3299 | 0.3717 |

Table 4: Zero-shot task-solving performances under various prompts for three open-source LLMs.

| Metrics | Flan-T5-Large | | Vicuna-7B | | LLaMA-2-13B | |
|---|---|---|---|---|---|---|
| | Prompt-1 | Prompt-2 | Prompt-1 | Prompt-2 | Prompt-1 | Prompt-2 |
| CLIP Score | 0.0 | 0.0 | 0.0 | 0.0 | 0.0 | 0.0 |
| BERT Score | 0.0 | 0.0 | 0.0603 | 0.0267 | 0.0971 | 0.1717 |
| ViT Score | 0.0 | 0.0 | 0.0 | 0.2385 | 0.0 | 0.0 |
| Overall | 0.0 | 0.0 | 0.0201 | 0.0884 | 0.0323 | 0.0572 |

GPT being instructed to generate a painting in a traditional Chinese style that depicts "Gao Shan Liu Shui". Initially, GPT seems to lack understanding of what constitutes a traditional Chinese style painting and it is also unfamiliar with the concept of "Gao Shan Liu Shui". As a remedy, GPT utilizes Google search in the initial two steps to gather information on these unfamiliar topics. Subsequently, it integrates the retrieved information to formulate a comprehensive prompt that instructs the Text-to-Image Generation model to create the desired artwork.

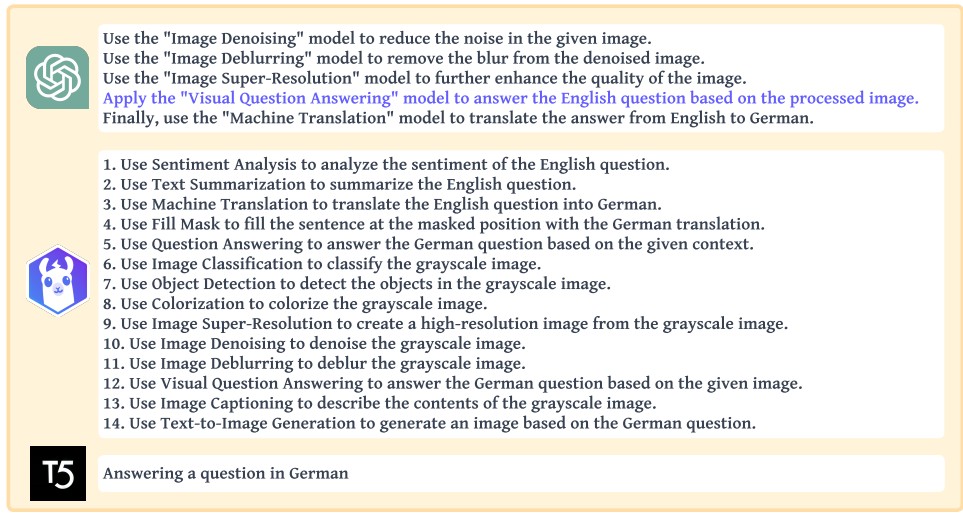

Figure 4: An example of Non-linear Planning.

## 6    Conclusions and Future Work

In this work, we introduce OpenAGI, an open-source AGI research platform designed to facilitate the development and evaluation of LLMs in solving complex, multi-step tasks through manipulating various domain expert models, tools, plugins or APIs. OpenAGI provides a wide range of tasks, models, datasets, benchmarks, and evaluation methods. We also propose the LLM+RLTF approach, which combines LLMs with reinforcement learning to optimize task-solving performance. The evaluation of various LLMs using the OpenAGI pipeline and different learning schema demonstrates that smaller-scale LLMs can potentially outperform larger models when combined with the appropriate learning approach, such as RLTF. In the future, we aim to explore 1) Human-in-the-loop agents, where LLM may prompt human experts for answers as one step of the task-solving plan when a suitable model is unavailable, thus enabling better Human-AI collaboration; 2) Trustworthy agents, which guarantee the safety and the ethical standard of agents during task-solving; and 3) Self-improving agents, which enable automated task generation and training that facilitate OpenAGI in independent exploration of tasks, empowering the self-reflection, self-prompting and self-improvement of intelligent agents.

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

# Supplementary Material for OpenAGI

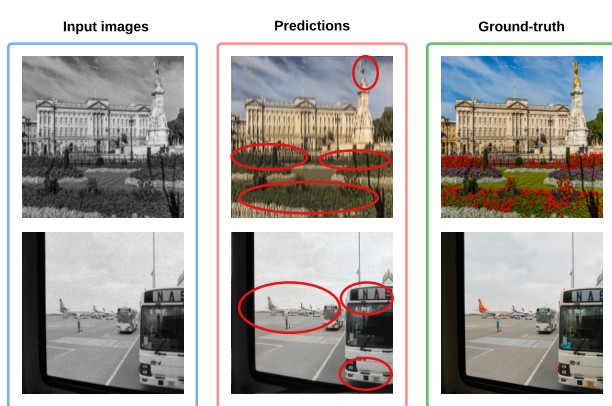

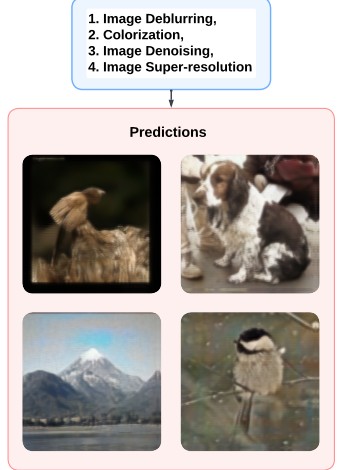

(a) Examples of the Out-of-Distribution Generalization issue for solving the same task (task description is the same as Fig. 1) with images from different distributions. The places highlighted by red ellipses denote areas with significant discrepancies from the ground-truth images after executing the same image restoration model sequence.

(b) Example of different model sequences for solving the same task depicted in Fig. 1. Both are valid model sequences but they result in very different task-solving quality.

Figure A.1: Research challenges when solving complex, multi-step tasks with augmented LLMs.

## A.1   Research Challenges

Although the OpenAGI platform offers numerous advantages and enhanced accessibility, it also gives rise to a variety of novel research challenges, such as:

- **Out-of-Distribution (OOD) Generalization.** Domain-specific expert models may exhibit limited generalization ability due to their strong dependence on the distribution of the training data. As demonstrated in Fig. A.1 (a), when processing images from disparate sources exhibiting a distributional shift, the original model sequence to address the task in Fig. 1 becomes ineffective. In the majority of instances, only a few colors are accurately restored, while most remain incorrect. Furthermore, noise and blurring persist, remaining highly perceptible to human observers.

- **Optimal Task Planning.** There is a compositional number of ways to combine different models to generate solutions, which can make it difficult to identify the best approach. Additionally, it is possible for multiple valid solutions to exist for a given task, but the quality of each solution can vary greatly. For instance, as depicted in Fig. A.1 (b), executing the same four models in a different sequence compared to Fig. 1 can lead to noticeably different outcomes. The results from the second approach (Fig. A.1 (b)) exhibit significantly more noise and color inconsistencies compared to the first approach (Fig. 1). Therefore, it is crucial for the LLM to identify and implement the optimal task plan from among the various possibilities.

- **Nonlinear Task Structures.** During model execution, a model may need more than one inputs and each input need to be produced by a prerequisite model, resulting in a nonlinear (tree) structure for the solution. In this context, employing a nonlinear task planning may enable more effective integration of the diverse inputs and more efficient parallel processing of the models to achieve the desired outcome. However, incorporating such nonlinear task planning ability into LLMs presents unique challenges beyond the LLM's existing task-solving capabilities.

In consideration of the first two challenges, we introduce a mechanism referred to as **Reinforcement Learning from Task Feedback (RLTF)**. This approach capitalizes on the performance feedback

procured from tasks following the execution of the solution devised by the LLM. Consequently, the RLTF mechanism effectively refines the LLM's planning strategy, resulting in an enhanced and more adaptive system. Indeed, relying solely on input text for learning proves insufficient for LLMs when confronted with real-world tasks. Task feedback, on the other hand, supplies additional information that steers the learning trajectory of LLMs towards improved and efficient solutions. For the third challenge, we propose **Nonlinear Task Planning**, which utilizes beam search as an efficient semi-autoregressive decoding method [11, 37] such that for each decoding step in beam search, different hypotheses are treated as parallel actionable solutions for different inputs instead of competing hypotheses. If a task requires parallel processing for multiple inputs, such as both text and image, then in generation time, an actionable solution taking text as input and another solution taking image as input will be generated and executed in parallel.

Table A.1: Language-related models

| Domain Task | Input Modality | Output Modality | Model |
|---|---|---|---|
| Sentiment Analysis | Text | Text | FinBert [3][1] |
| Text Summarization | Text | Text | BART [4][20] |
| Machine Translation | Text | Text | T5 [5][34] |
| Fill Mask | Text | Text | DistilRoberta [6][22] |
| Question Answering | Text, Text | Text | DistilBERT [7][39] |

Table A.2: Vision-related models

| Domain Task | Input Modality | Output Modality | Model |
|---|---|---|---|
| Image Classification | Image | Text | ViT [8][12] |
| Object Detection | Image | Text | DETR [9][3] |
| Colorization | Image | Image | Colorizer [10][54] |
| Image Super-Resolution | Image | Image | Swin2SR [11][10] |
| Image Denoising | Image | Image | Restormer [12][53] |
| Image Deblurring | Image | Image | Restormer [53] |

Table A.3: Vision-language models

| Domain Task | Input Modality | Output Modality | Model |
|---|---|---|---|
| Visual Question Answering | Image, Text | Text | GIT [13][48] |
| Image Captioning | Image | Text | Vision Encoder Decoder [14] |
| Text-to-Image Generation | Text | Image | Stable Diffusion [15][36] |

[1]https://huggingface.co/yiyanghkust/finbert-tone

[2]https://huggingface.co/distilbert-base-cased-distilled-squad

[3]https://huggingface.co/facebook/bart-large-cnn

[4]https://huggingface.co/gpt2

[5]https://huggingface.co/t5-base

[6]https://huggingface.co/distilroberta-base

[7]https://huggingface.co/google/vit-base-patch16-224

[8]https://huggingface.co/facebook/detr-resnet-101

[9]https://github.com/richzhang/colorization

[10]https://huggingface.co/caidas/swin2SR-classical-sr-x2-64

[11]https://github.com/swz30/Restormer

[12]https://huggingface.co/microsoft/git-base-textvqa

[13]https://huggingface.co/nlpconnect/vit-gpt2-image-captioning

[14]https://huggingface.co/CompVis/stable-diffusion-v1-4

## A.2   Original Datasets

- **ImageNet-1K** [38] is a large-scale image dataset, derived from the broader ImageNet database, containing approximately 1 million images. These images are categorized into 1,000 distinct classes, with each class representing a specific object or concept. The dataset has been instrumental in the development and evaluation of state-of-the-art deep learning algorithms for image classification, object recognition, and transfer learning.

- **Common Objects in Context (COCO)** [21] is a large-scale, richly-annotated image dataset designed to advance the fields of object detection, segmentation, and captioning. Released in 2014, it contains over 200,000 labeled images with 1.5 million object instances from 80 different object categories. The dataset features complex, real-world scenes with multiple objects per image, various object scales, and diverse contexts.

- **CNN/Daily Mail** [26] is a valuable resource for text summarization, which consists of human-generated abstractive summaries, created by transforming news articles from CNN and Daily Mail websites into questions, with one entity concealed, and generating summaries from the corresponding passages. The authors have made available the scripts used to crawl, extract, and generate question-answer pairs from these websites. The corpus contains 286,817 training pairs, 13,368 validation pairs, and 11,487 test pairs, as defined by the scripts. On average, the source documents in the training set span 766 words across 29.74 sentences, while the summaries are composed of 53 words and 3.72 sentences.

- **Stanford Sentiment Treebank (SST2)** [28] is a corpus with labeled parse trees that allows for the analysis of the compositional effects of sentiment in language. The corpus consists of 11,855 single sentences extracted from movie reviews. It was parsed with the Stanford parser and includes a total of 215,154 unique phrases from those parse trees, each annotated by 3 human judges.

- **TextVQA** [43] serves as a benchmark for evaluating visual reasoning based on text present in images. In order to answer questions pertaining to the images, TextVQA necessitates models to read and reason about the text contained within them. The incorporation of text as a new modality in images demands that models be able to reason over this modality to address TextVQA queries. Thus, TextVQA poses a unique challenge for models to integrate both visual and textual cues to arrive at a comprehensive answer.

- **Stanford Question Answering Dataset (SQuAD)** [35] is a collection of question-answer pairs sourced from Wikipedia articles. A distinguishing characteristic of SQuAD is that the correct answers to the questions can be any sequence of tokens in the corresponding text. This flexibility is a result of the dataset's construction through crowd-sourcing, which results in a diverse set of questions and answers compared to other question-answering datasets.

## A.3   Data Augmentation Methods

Upon determining the raw datasets, our next objective is to augment them from various perspectives to construct complex, multi-step tasks. For instance, we can introduce noise and reduce the resolution of an image from ImageNet-1K to create new datasets that may require "Image Denoising" and "Image Super-Resolution" for initial recovery before performing classification. The data augmentation methods employed are as follows:

- **Gaussian Blur** is a prevalent image processing technique that involves convolving an image with a Gaussian filter kernel. This filter is applied to smooth the image and reduce noise, yielding a blurred output image.

- **Gaussian Noise** refers to the addition of Gaussian-distributed noise.

- **Grayscale** entails converting the colorful image to a grayscale image.

- **Low Resolution** pertains to images with a reduced pixel density (pixels per inch, or ppi).

- **Translation** denotes the process of converting a text from one language, such as English, to another, such as German. In this work, we only use English-to-German translator for simplicity.

- **Word Mask** randomly replaces a single word in a given sentence with the "[MASK]" token.

| Task description | Input Sample | Output Sample |
|---|---|---|
| Given low-resolutioned noisy blurry grayscale image, how to return the regular image step by step? |  |  |
| Given low-resolution noisy blurry grayscale image, how to return the object names in English step by step? |  | bear |
| Given clozed English text, how to translate the text in German step by step? | A big burly grizzly bear is show [Mask] grass in the background. | Ein kräftiger Grizzly Bär ist im Hintergrund mit Gras zu sehen. |
| Given noisy blurry grayscale image and clozed English query, how to answer the question in English step by step? |  **Question**: what number is [Mask] the player's jersey? | 22 |
| Given clozed English document and clozed English query, how to answer the question in German step by step? | **Context**: Super Bowl 5 was an American football game to determine the champion of the National... **Question**: What was the theme of Super [Mask] 50? | Goldener Jahrestag |

Table A.4: Examples of multi-step tasks and their augmented data samples.

## A.4   Evaluation Metrics

- **CLIP Score**[16] is a reference-free metric used to assess the correlation between a generated image caption and the actual content of the image. Research has shown that it has a strong correlation with human judgment and is a reliable measure for evaluating image captioning performance [16].

- **BERT Score**[17] uses contextual embeddings from the pre-trained BERT model to compare words in candidate and reference sentences through cosine similarity. Studies have shown that it is highly correlated with human evaluation at both sentence-level and system-level [56]. Additionally, BERT Score calculates precision, recall, and F1 measure, making it a valuable tool for evaluating various language generation tasks. In this work, we use the value of F1 score.

- **ViT Score**[18] is a metric designed to assess the visual similarity between two images. By calculating the cosine similarity of their respective embeddings, which are generated using a Vision Transformer, the ViT Score offers a quantitative measure of their likeness.

---

[16]https://torchmetrics.readthedocs.io/en/stable/multimodal/clip_score.html

[17]https://huggingface.co/spaces/evaluate-metric/bertscore

[18]https://colab.research.google.com/github/huggingface/notebooks/blob/main/examples/image_similarity.ipynb

## A.5  Dataset Documentation and Data Samples for Benchmark Tasks

Our dataset is designed to evaluate LLM's planning ability of using domain expert models. To accomplish this, we enhance the standard CV/NLP datasets using various combinations of data augmentation methodologies. We have devised 185 multi-step tasks in total, of which 117 tasks maintain a linear task structure with steps following a simple sequence, while the remaining 68 tasks exhibit a non-linear task structure, where steps might be performed concurrently or in a complex order. Each benchmark task is accompanied by a small dataset, which contains 100 augmented data samples. All benchmark datasets can be accessed, reviewed, and downloaded via `https://drive.google.com/drive/folders/1AjT6y7qLIMxcmHhUBG5IE1_5SnCPR57e`, which is committed to transparency and ease of accessibility. As the authors, we affirm that we assume all responsibility for any rights violation related to this dataset. The data license is **Creative Commons Attribution 4.0 International**, ensuring all necessary permissions and regulations are stringently adhered to. The dataset is hosted on GitHub `https://github.com/agiresearch/OpenAGI`. We have chosen this platform considering its robustness, reliability, and its proven track record for data hosting. We ensure that access to the data will be maintained consistently, possibly through a curated interface. A maintenance plan is in place to address potential issues, provide necessary updates, and ensure the data's long-term availability and integrity.

We also offer several data samples to illustrate the structure of the datasets further. For example, consider the third row of Tab. A.4, which represents a machine translation domain task (i.e., translating from English to German). In this case, we apply the "Word Mask" augmentation technique on the text inputs to create a multi-step task, which can be described as "Given clozed English text, how can the text be translated into German step by step?" For instance, given an original data sample, "A big burly grizzly bear is shown with grass in the background", the word "with" has been chosen to be masked to generate the augmented data sample, "A big burly grizzly bear is shown [MASK] grass in the background".

## A.6  Details of RLTF

In the setup of RLTF, the environment is the OpenAGI platform and the agent is the LLM $\mathcal{L}$ parameterized with $\Phi$. The solution $s$ generated by the LLM can be seen as a set of instructions that solve the input task $t$ and can be executed on the corresponding augmented dataset $\mathcal{D}_t$. We can use the performance (provided in Sec. 3.1.3) on that dataset as the reward signal $\mathcal{R}$ and use reinforcement learning to fine-tune the LLM. More concretely, to find the optimal solution, we require the LLM to maximize its expected reward on the training set $\mathcal{T}_{train}$, represented by $J(\Phi)$:

$$J(\Phi) = \mathbb{E}_{\mathbf{s}_{train} \sim \mathcal{L}(\mathcal{T}_{train}|\Phi)}[\mathcal{R}] \tag{A.1}$$

Since the reward signal $\mathcal{R}$ is non-differentiable, we need to use a policy gradient method to iteratively update $\Phi$. In this work, we use the REINFORCE in [50] as follows,

$$\nabla_\Phi J(\Phi) = \mathbb{E}_{P(\mathbf{s}_{train}|\Phi)}[\nabla_\Phi \log P(\mathbf{s}_{train}|\Phi) \cdot \mathcal{R}] \tag{A.2}$$

An empirical approximation of the above quantity is:

$$\nabla_\Phi J(\Phi) \approx \frac{1}{|\mathcal{T}_{train}|} \sum_{t \in \mathcal{T}_{train}} \nabla_\Phi \log P(s_{train}|\Phi) \cdot \mathcal{R} \tag{A.3}$$

The above update is an unbiased estimate for our gradient, but has a very high variance. To reduce the variance of this estimate, we employ a baseline function $b$, which is the moving average of the previous reward signals:

$$\nabla_\Phi J(\Phi) \approx \frac{1}{|\mathcal{T}_{train}|} \sum_{t \in \mathcal{T}_{train}} \nabla_\Phi \log P(s_{train}|\Phi) \cdot (\mathcal{R} - b) \tag{A.4}$$

## A.7  Constrained Generation

To generate the solution for a natural language task description, we require the LLM to generate an actionable solution consisting of sequences of model names. For tasks that require only one input,

the model only needs to generate one actionable sequence of models. For tasks that require multiple inputs, such as Visual Question Answering, the LLM needs multiple steps in order to accomplish the task, where each step is either a sequence of models or a parallel of several sequences of models. Towards this end, the LLM must satisfy three conditions: 1) only generate the model names without irrelevant tokens, 2) generate valid sequences of models, and 3) generate paralleled sequences of models for different inputs when necessary.

**Condition 1**: For the LLM to generate only model names, instead of tuning the model to teach it what names are available, we adopt constrained beam search [11], which only allows generating tokens from the model set $\mathcal{M}$ at every decoding step. More specifically, we define our constraints as a prefix trie such that each model name is a path from the root to some leaf node. For each node $t$ in the tree, its children indicate all the allowed continuations from the prefix defined traversing the trie from the root to $t$. Thus in each decoding step, the next token can only be selected from either all possible continuations allowed based on the generated tokens or the first tokens of all possible next model names. For example, if "Text" is already generated, based on the set of model names, the next tokens can only be either "Summarization" due to the "Text Summarization" model or "Generation" due to the "Text Generation" model, as shown in Fig. A.2.

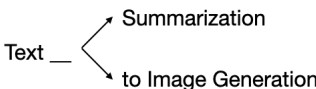

Figure A.2: Model name based constrained generation.

**Condition 2**: For the LLM to generate valid sequences of models, consecutive models should have input and output modalities matched. If the output modality of a model is text, then the next model can only be models that take text as input. This is also achieved by constrained beam search such that when finishing generating one model, the constraint function will determine the output modality of this model and find out all possible next models in model set $\mathcal{M}$, excluding the models that are already generated. It will dynamically construct a new trie for all these model names based on the output modality. For example, if the first generated model name is "Text Summarization", then the next possible models can be "Sentiment Analysis", "Text Generation", etc., as shown in Fig. A.3.

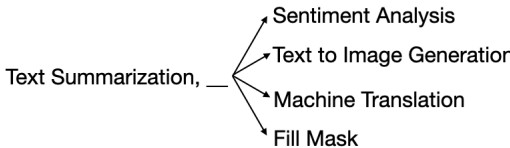

Figure A.3: Model type based constrained generation.

## A.8   Zero- and Few-shot Schema

As in shown in Fig. A.4, in the zero-shot setting, most LLMs struggle to generate valid task plans, let alone optimal solutions. In particular, GPT-3.5 tends to generate repetitive contents, which subsequently maps to identical model names. Meanwhile, Vicuna-7B and Flan-T5-Large, constrained by their zero-shot capabilities, fail to produce a reasonable plan. In the few-shot setting, we incorporate several manually labeled task plans as instructions to guide the generation, resulting in a remarkable improvement in the quality of the task plans. As observed in Fig. A.5, all three LLMs can produce solutions that are semantically similar to the provided examples. In fact, many solutions can be used directly, even without the need for mapping.

## A.9   Broader Impacts, Limitations, and Future Visions

Just like any technology, the irresponsible use of AI techniques and intelligent systems may have detrimental effects on individuals and society as a whole. In particular, existing Large Language Models (LLMs) are not sufficiently designed to ensure their harmless usage, making them vulnerable to misuse by malicious parties. Consequently, it is important to address and mitigate the potential risks associated with LLMs when used for complex task solving. Our constrained generation framework provides a potential solution to this issue. By incorporating ethical constraints, such as an AI

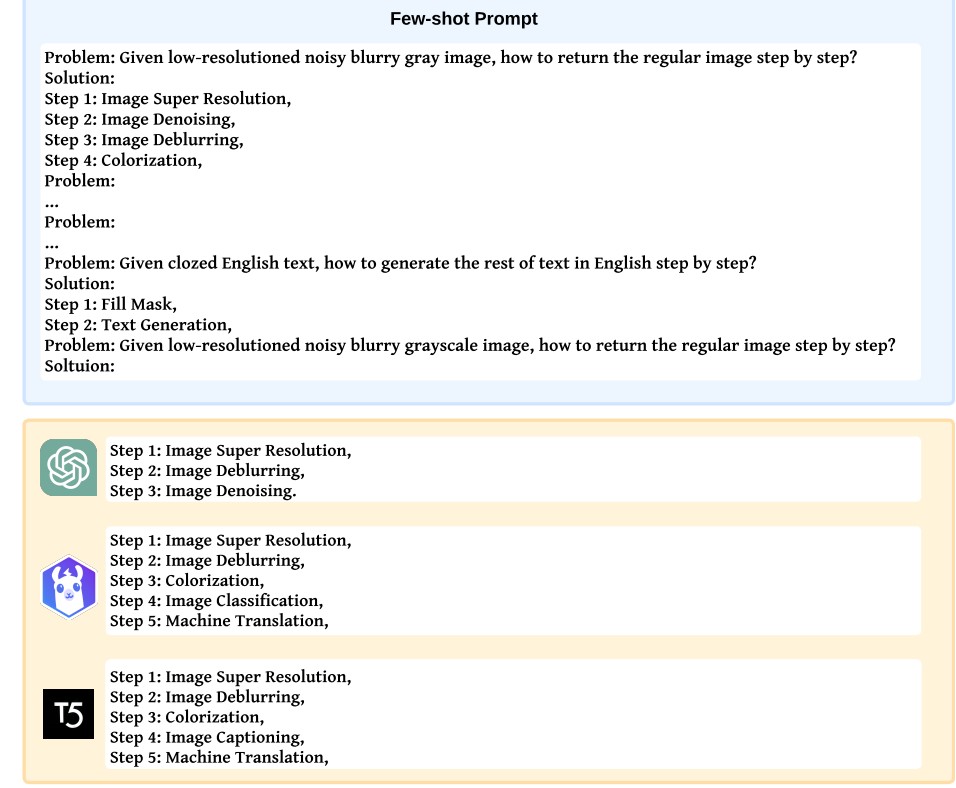

**Zero-shot Prompt**

Given a low-resolution, blurry, grayscale image, how to return the regular image step by step?

Step 1: Collect more data or images to create a higher resolution version of the blurry image.
Step 2: Use a deblurring algorithm to remove the blurriness and increase the sharpness of the image.
Step 3: Apply a denoising algorithm to reduce any noise in the image that may have been introduced during the deblurring process.
Step 4: Use an upsampling method to increase the resolution of the image to the desired level.
Step 5: Apply a contrast enhancement technique to make the image more visually appealing.
Step 6: Finally, apply sharpening techniques to add crispness and detail to the image.

1. Use the image to generate the mask image.
2. Use the mask image to generate the masked image.
3. Use the masked image to generate the final image.
4. Use the masked image to generate the final image.
5. Use ...
...

The blurry image is a blurry image with a blurry background. To return the regular image, you can use the following steps: First, you need to find the blurry image. Then, you need to find the regular image. Then, you need to find the blurry image. Then, you need to find the regular image. Then, you need to find the blurry image. Then, you need to find the regular image. Then, ...

Figure A.4: An example of zero-shot schema.

**Few-shot Prompt**

Problem: Given low-resolutioned noisy blurry gray image, how to return the regular image step by step?
Solution:
Step 1: Image Super Resolution,
Step 2: Image Denoising,
Step 3: Image Deblurring,
Step 4: Colorization,
Problem:
...
Problem:
...
Problem: Given clozed English text, how to generate the rest of text in English step by step?
Solution:
Step 1: Fill Mask,
Step 2: Text Generation,
Problem: Given low-resolutioned noisy blurry grayscale image, how to return the regular image step by step?
Soltuion:

Step 1: Image Super Resolution,
Step 2: Image Deblurring,
Step 3: Image Denoising.

Step 1: Image Super Resolution,
Step 2: Image Deblurring,
Step 3: Colorization,
Step 4: Image Classification,
Step 5: Machine Translation,

Step 1: Image Super Resolution,
Step 2: Image Deblurring,
Step 3: Colorization,
Step 4: Image Captioning,
Step 5: Machine Translation,

Figure A.5: An example of few-shot schema.

constitution, into the plan generation process, we can guide the agent to generate plans that are both ethically sound and benign while tackling complex tasks.

In the future, we can incorporate multiple models within each single-step task, thereby providing an expanded selection of options for LLMs to address complex problems. Additionally, we can integrate datasets from alternative modalities, such as video and audio, into the OpenAGI platform. These datasets will facilitate the development of more sophisticated tasks to further investigate the planning capabilities of LLMs. We will also explore multiple task-solving agents to interact with each other for complex problem solving. Another promising direction is to involve humans in the loop during the

resolution of complex tasks. In such scenarios, LLM may prompt human experts for answers as one step of the task-solving plan when a suitable model is unavailable, thus enabling better Human-AI collaboration. Lastly, we can explore automated task generation techniques that empower OpenAGI to generate complex tasks independently, facilitating self-prompting and self-improvement in its task-solving capabilities.

## A.10    Computational Resources

For augmenting the data, we used devices equipped with Intel(R) Xeon(R) Gold 6226R CPU @ 2.90GHz and 256 GB RAM. For training and testing the LLMs, we used 4xA5000-24GB GPUs.

## A.11    Training Details

In our experiments, we use Low-Rank Adaptation (LoRA)[19] [17] for efficient fine-tuning of Flan-T5-Large, Vicuna-7B and LLaMA-2-13B under both Fine-tuning and RLTF schema, and the configuration/hyper-parameter settings are shown in Tab. A.5.

Table A.5: Configuration and parameter settings for Flan-T5-Large, Vicuna-7B and LLaMA-2-13B

| Configuration/Hyper-parameter | Flan-T5-Large | | Vicuna-7B | | LLaMA-2-13B | |
| --- | --- | --- | --- | --- | --- | --- |
| | Fine-tuning | RLTF | Fine-tuning | RLTF | Fine-tuning | RLTF |
| Optimizer | AdamW | AdamW | AdamW | AdamW | AdamW | AdamW |
| Epochs | 200 | 10 | 200 | 10 | 200 | 10 |
| Training Batch Size Per GPU | 8 | 5 | 1 | 1 | 1 | 1 |
| Gradient Accumulation Steps | 1 | 1 | 1 | 1 | 1 | 1 |
| Learning Rate | 1e-5 | 1e-5 | 5e-6 | 5e-6 | 1e-5 | 1e-5 |
| Weight Decay | 1e-6 | 1e-6 | 1e-6 | 1e-6 | 1e-6 | 1e-6 |
| Warmup Ratio | 0.1 | 0.1 | 0 | 0 | 0 | 0 |
| Scheduler | Linear | Linear | Linear | Linear | Linear | Linear |
| LoRA_r | 8 | 8 | 8 | 8 | 8 | 8 |
| LoRA_$\alpha$ | 16 | 16 | 16 | 16 | 16 | 16 |
| LoRA_dropout | 0.05 | 0.05 | 0.05 | 0.05 | 0.05 | 0.05 |
| $\epsilon$ | - | 0.2 | - | 0.2 | - | 0.2 |
| Decay Rate of $\epsilon$ | - | 0.9 | - | 0.9 | - | 0.9 |
| Beam Size | - | 30 | - | 20 | - | 20 |
| Num of Outputs | - | 30 | - | 20 | - | 20 |
| Top k | - | 5 | - | 40 | - | 40 |
| Top p | - | 0.5 | - | 0.75 | - | 0.75 |
| Temperature | - | 0.9 | - | 0.2 | - | 0.2 |
| Num of Beam Groups | - | 1 | - | 1 | - | 1 |

Table A.6: Task descriptions of all multi-step tasks in OpenAGI. The difficulty level is estimated by the size of human-labeled solutions, that is, the total number of models used in the human-labeled task solution. The higher the number, the more difficult the task.

| Task Description | Difficulty Level |
| --- | --- |
| Given low-resolutioned noisy blurry grayscale image how to return the regular image step by step? | 4 |
| Given noisy blurry grayscale image how to return the regular image step by step? | 3 |
| Given low-resolutioned blurry grayscale image how to return the regular image step by step? | 3 |
| Given blurry grayscale image how to return the regular image step by step? | 2 |
| Given low-resolutioned noisy grayscale image how to return the regular image step by step? | 3 |
| | Continued on next page |

---

[19]https://huggingface.co/blog/lora

| Task Description | Difficulty Level |
| --- | --- |
| Given noisy grayscale image
how to return the regular image step by step? | 2 |
| Given low-resolutioned grayscale image
how to return the regular image step by step? | 2 |
| Given grayscale image
how to return the regular image step by step? | 1 |
| Given low-resolutioned noisy blurry image
how to return the regular image step by step? | 3 |
| Given noisy blurry image
how to return the regular image step by step? | 2 |
| Given low-resolutioned blurry image
how to return the regular image step by step? | 2 |
| Given blurry image
how to return the regular image step by step? | 1 |
| Given low-resolutioned noisy image
how to return the regular image step by step? | 2 |
| Given noisy image
how to return the regular image step by step? | 1 |
| Given low-resolutioned image
how to return the regular image step by step? | 1 |
| Given low-resolutioned noisy blurry grayscale image
how to return the caption in German step by step? | 5 |
| Given low-resolutioned noisy blurry grayscale image
how to return the class label in German step by step? | 6 |
| Given low-resolutioned noisy blurry grayscale image
how to return the object names in German step by step? | 6 |
| Given low-resolutioned noisy blurry grayscale image
how to return the caption in English step by step? | 5 |
| Given low-resolutioned noisy blurry grayscale image
how to return the class label in English step by step? | 5 |
| Given low-resolutioned noisy blurry grayscale image
how to return the object names in English step by step? | 5 |
| Given noisy blurry grayscale image
how to return the caption in German step by step? | 5 |
| Given noisy blurry grayscale image
how to return the class label in German step by step? | 5 |
| Given noisy blurry grayscale image
how to return the object names in German step by step? | 5 |
| Given noisy blurry grayscale image
how to return the caption in English step by step? | 4 |
| Given noisy blurry grayscale image
how to return the class label in English step by step? | 4 |
| Given noisy blurry grayscale image
how to return the object names in English step by step? | 4 |
| Given low-resolutioned blurry grayscale image
how to return the caption in German step by step? | 5 |
| Given low-resolutioned blurry grayscale image
how to return the class label in German step by step? | 5 |
| Given low-resolutioned blurry grayscale image
how to return the object names in German step by step? | 5 |
| Given low-resolutioned blurry grayscale image
how to return the caption in English step by step? | 4 |
| Given low-resolutioned blurry grayscale image
how to return the class label in English step by step? | 4 |
| Given low-resolutioned blurry grayscale image
how to return the object names in English step by step? | 4 |
| Given blurry grayscale image
how to return the caption in German step by step? | 4 |
| Given blurry grayscale image
how to return the class label in German step by step? | 4 |
| | Continued on next page |

| Task Description | Difficulty Level |
|---|---|
| Given blurry grayscale image how to return the object names in German step by step? | 4 |
| Given blurry grayscale image how to return the caption in English step by step? | 3 |
| Given blurry grayscale image how to return the class label in English step by step? | 3 |
| Given blurry grayscale image how to return the object names in English step by step? | 3 |
| Given low-resolutioned noisy grayscale image how to return the caption in German step by step? | 5 |
| Given low-resolutioned noisy grayscale image how to return the class label in German step by step? | 5 |
| Given low-resolutioned noisy grayscale image how to return the object names in German step by step? | 5 |
| Given low-resolutioned noisy grayscale image how to return the caption in English step by step? | 4 |
| Given low-resolutioned noisy grayscale image how to return the class label in English step by step? | 4 |
| Given low-resolutioned noisy grayscale image how to return the object names in English step by step? | 4 |
| Given noisy grayscale image how to return the caption in German step by step? | 4 |
| Given noisy grayscale image how to return the class label in German step by step? | 4 |
| Given noisy grayscale image how to return the object names in German step by step? | 4 |
| Given noisy grayscale image how to return the caption in English step by step? | 3 |
| Given noisy grayscale image how to return the class label in English step by step? | 3 |
| Given noisy grayscale image how to return the object names in English step by step? | 3 |
| Given low-resolutioned grayscale image how to return the caption in German step by step? | 4 |
| Given low-resolutioned grayscale image how to return the class label in German step by step? | 4 |
| Given low-resolutioned grayscale image how to return the object names in German step by step? | 4 |
| Given low-resolutioned grayscale image how to return the caption in English step by step? | 3 |
| Given low-resolutioned grayscale image how to return the class label in English step by step? | 3 |
| Given low-resolutioned grayscale image how to return the object names in English step by step? | 3 |
| Given grayscale image how to return the caption in German step by step? | 3 |
| Given grayscale image how to return the class label in German step by step? | 3 |
| Given grayscale image how to return the object names in German step by step? | 3 |
| Given grayscale image how to return the caption in English step by step? | 2 |
| Given grayscale image how to return the class label in English step by step? | 2 |
| Given grayscale image how to return the object names in English step by step? | 2 |
| Given low-resolutioned noisy blurry image how to return the caption in German step by step? | 5 |
| Given low-resolutioned noisy blurry image how to return the class label in German step by step? | 5 |

| Task Description | Difficulty Level |
|---|---|
| Given low-resolutioned noisy blurry image
how to return the object names in German step by step? | 5 |
| Given low-resolutioned noisy blurry image
how to return the caption in English step by step? | 4 |
| Given low-resolutioned noisy blurry image
how to return the class label in English step by step? | 4 |
| Given low-resolutioned noisy blurry image
how to return the object names in English step by step? | 4 |
| Given noisy blurry image
how to return the caption in German step by step? | 4 |
| Given noisy blurry image
how to return the class label in German step by step? | 4 |
| Given noisy blurry image
how to return the object names in German step by step? | 4 |
| Given noisy blurry image
how to return the caption in English step by step? | 3 |
| Given noisy blurry image
how to return the class label in English step by step? | 3 |
| Given noisy blurry image
how to return the object names in English step by step? | 3 |
| Given low-resolutioned blurry image
how to return the caption in German step by step? | 4 |
| Given low-resolutioned blurry image
how to return the class label in German step by step? | 4 |
| Given low-resolutioned blurry image
how to return the object names in German step by step? | 4 |
| Given low-resolutioned blurry image
how to return the caption in English step by step? | 3 |
| Given low-resolutioned blurry image
how to return the class label in English step by step? | 3 |
| Given low-resolutioned blurry image
how to return the object names in English step by step? | 3 |
| Given blurry image
how to return the caption in German step by step? | 3 |
| Given blurry image
how to return the class label in German step by step? | 3 |
| Given blurry image
how to return the object names in German step by step? | 3 |
| Given blurry image
how to return the caption in English step by step? | 2 |
| Given blurry image
how to return the class label in English step by step? | 2 |
| Given blurry image
how to return the object names in English step by step? | 2 |
| Given low-resolutioned noisy image
how to return the caption in German step by step? | 4 |
| Given low-resolutioned noisy image
how to return the class label in German step by step? | 4 |
| Given low-resolutioned noisy image
how to return the object names in German step by step? | 4 |
| Given low-resolutioned noisy image
how to return the caption in English step by step? | 3 |
| Given low-resolutioned noisy image
how to return the class label in English step by step? | 3 |
| Given low-resolutioned noisy image
how to return the object names in English step by step? | 3 |
| Given noisy image
how to return the caption in German step by step? | 3 |
| Given noisy image
how to return the class label in German step by step? | 3 |

| Task Description | Difficulty Level |
|---|---|
| Given noisy image
how to return the object names in German step by step? | 3 |
| Given noisy image
how to return the caption in English step by step? | 2 |
| Given noisy image
how to return the class label in English step by step? | 2 |
| Given noisy image
how to return the object names in English step by step? | 2 |
| Given low-resolutioned image
how to return the caption in German step by step? | 3 |
| Given low-resolutioned image
how to return the class label in German step by step? | 3 |
| Given low-resolutioned image
how to return the object names in German step by step? | 3 |
| Given low-resolutioned image
how to return the caption in English step by step? | 2 |
| Given low-resolutioned image
how to return the class label in English step by step? | 2 |
| Given low-resolutioned image
how to return the object names in English step by step? | 2 |
| Given clozed English text
how to generate a image step by step? | 2 |
| Given English text
how to generate a image step by step? | 1 |
| Given clozed English text
how to return the summarization in German step by step? | 3 |
| Given clozed English text
how to translate the text in German step by step? | 2 |
| Given clozed English text
how to return the sentiment in German step by step? | 3 |
| Given clozed English text
how to return the summarization in English step by step? | 2 |
| Given clozed English text
how to return the sentiment in English step by step? | 2 |
| Given English text
how to return the summarization in German step by step? | 2 |
| Given English text
how to translate the text in German step by step? | 1 |
| Given English text
how to return the sentiment in German step by step? | 2 |
| Given English text
how to return the summarization in English step by step? | 1 |
| Given English text
how to return the sentiment in English step by step? | 1 |
| Given low-resolutioned noisy blurry grayscale image and clozed English query
how to answer the question in English step by step? | 6 |
| Given low-resolutioned noisy blurry grayscale image and clozed English query
how to answer the question in German step by step? | 7 |
| Given low-resolutioned noisy blurry grayscale image and English query
how to answer the question in English step by step? | 5 |
| Given low-resolutioned noisy blurry grayscale image and English query
how to answer the question in German step by step? | 6 |
| Given noisy blurry grayscale image and clozed English query
how to answer the question in English step by step? | 5 |
| Given noisy blurry grayscale image and clozed English query
how to answer the question in German step by step? | 6 |
| Given noisy blurry grayscale image and English query
how to answer the question in English step by step? | 4 |
| Given noisy blurry grayscale image and English query
how to answer the question in German step by step? | 5 |
| | Continued on next page |

| Task Description | Difficulty Level |
|---|---|
| Given low-resolutioned blurry grayscale image and clozed English query how to answer the question in English step by step? | 5 |
| Given low-resolutioned blurry grayscale image and clozed English query how to answer the question in German step by step? | 6 |
| Given low-resolutioned blurry grayscale image and English query how to answer the question in English step by step? | 4 |
| Given low-resolutioned blurry grayscale image and English query how to answer the question in German step by step? | 5 |
| Given blurry grayscale image and clozed English query how to answer the question in English step by step? | 4 |
| Given blurry grayscale image and clozed English query how to answer the question in German step by step? | 5 |
| Given blurry grayscale image and English query how to answer the question in English step by step? | 3 |
| Given blurry grayscale image and English query how to answer the question in German step by step? | 4 |
| Given low-resolutioned noisy grayscale image and clozed English query how to answer the question in English step by step? | 4 |
| Given low-resolutioned noisy grayscale image and clozed English query how to answer the question in German step by step? | 6 |
| Given low-resolutioned noisy grayscale image and English query how to answer the question in English step by step? | 4 |
| Given low-resolutioned noisy grayscale image and English query how to answer the question in German step by step? | 5 |
| Given noisy grayscale image and clozed English query how to answer the question in English step by step? | 5 |
| Given noisy grayscale image and clozed English query how to answer the question in German step by step? | 5 |
| Given noisy grayscale image and English query how to answer the question in English step by step? | 3 |
| Given noisy grayscale image and English query how to answer the question in German step by step? | 4 |
| Given low-resolutioned grayscale image and clozed English query how to answer the question in English step by step? | 4 |
| Given low-resolutioned grayscale image and clozed English query how to answer the question in German step by step? | 5 |
| Given low-resolutioned grayscale image and English query how to answer the question in English step by step? | 3 |
| Given low-resolutioned grayscale image and English query how to answer the question in German step by step? | 4 |
| Given grayscale image and clozed English query how to answer the question in English step by step? | 4 |
| Given grayscale image and clozed English query how to answer the question in German step by step? | 5 |
| Given grayscale image and English query how to answer the question in English step by step? | 2 |
| Given grayscale image and English query how to answer the question in German step by step? | 3 |
| Given low-resolutioned noisy blurry image and clozed English query how to answer the question in English step by step? | 4 |
| Given low-resolutioned noisy blurry image and clozed English query how to answer the question in German step by step? | 5 |
| Given low-resolutioned noisy blurry image and English query how to answer the question in English step by step? | 4 |
| Given low-resolutioned noisy blurry image and English query how to answer the question in German step by step? | 5 |
| Given noisy blurry image and clozed English query how to answer the question in English step by step? | 4 |
| Given noisy blurry image and clozed English query how to answer the question in German step by step? | 5 |

| Task Description | Difficulty Level |
|---|---|
| Given noisy blurry image and English query how to answer the question in English step by step? | 3 |
| Given noisy blurry image and English query how to answer the question in German step by step? | 4 |
| Given low-resolutioned blurry image and clozed English query how to answer the question in English step by step? | 4 |
| Given low-resolutioned blurry image and clozed English query how to answer the question in German step by step? | 5 |
| Given low-resolutioned blurry image and English query how to answer the question in English step by step? | 3 |
| Given low-resolutioned blurry image and English query how to answer the question in German step by step? | 4 |
| Given blurry image and clozed English query how to answer the question in English step by step? | 3 |
| Given blurry image and clozed English query how to answer the question in German step by step? | 4 |
| Given blurry image and English query how to answer the question in English step by step? | 2 |
| Given blurry image and English query how to answer the question in German step by step? | 3 |
| Given low-resolutioned noisy image and clozed English query how to answer the question in English step by step? | 4 |
| Given low-resolutioned noisy image and clozed English query how to answer the question in German step by step? | 5 |
| Given low-resolutioned noisy image and English query how to answer the question in English step by step? | 3 |
| Given low-resolutioned noisy image and English query how to answer the question in German step by step? | 4 |
| Given noisy image and clozed English query how to answer the question in English step by step? | 3 |
| Given noisy image and clozed English query how to answer the question in German step by step? | 4 |
| Given noisy image and English query how to answer the question in English step by step? | 3 |
| Given noisy image and English query how to answer the question in German step by step? | 4 |
| Given low-resolutioned image and clozed English query how to answer the question in English step by step? | 3 |
| Given low-resolutioned image and clozed English query how to answer the question in German step by step? | 4 |
| Given low-resolutioned image and English query how to answer the question in English step by step? | 2 |
| Given low-resolutioned image and English query how to answer the question in German step by step? | 3 |
| Given clozed English document and clozed English query how to answer the question in German step by step? | 4 |
| Given clozed English document and clozed English query how to answer the question in English step by step? | 3 |
| Given clozed English document and English query how to answer the question in German step by step? | 3 |
| Given clozed English document and English query how to answer the question in English step by step? | 2 |
| Given English document and clozed English query how to answer the question in German step by step? | 3 |
| Given English document and clozed English query how to answer the question in English step by step? | 2 |
| Given English document and English query how to answer the question in German step by step? | 2 |
| Given English document and English query how to answer the question in English step by step? | 1 |

**Prompt-1**

You are a planner who is an expert at coming up with a to-do list for a given objective for the execution of a robot. Ensure the list is as short as possible. Each task in it is relevant, effective, short and necessary. The robot is only allowed to utilize the provided machine learning models to do each task. Develop a to-do list to achieve the objective: Given a noisy, blurry, grayscale image and English question related to that image, how to answer the question in German?

Provided models:
Sentiment Analysis
Text Summarization
Machine Translation
Fill Mask
Question Answering
Image Classification
Object Detection
Colorization
Image Super-Resolution
Image Denoising
Image Deblurring
Visual Question Answering
Image Captioning
Text-to-Image Generation

---

**Prompt-2**

You are a planner who is an expert at coming up with a to-do list for a given objective for the execution of a robot. Ensure the list is as short as possible. Each task in it is relevant, effective, short and necessary. The robot is only allowed to utilize the provided machine learning models to do each task. Develop a to-do list to achieve the objective: Given a noisy, blurry, grayscale image and English question related to that image, how to answer the question in German?

Provided models:
Sentiment Analysis: useful when you want to analyze the sentiment of a sentence. It receives sentence as input.
Text Summarization: useful when you want to summarize a sentence or a paragraph. It receives text as input.
Machine Translation: useful when you want to translate a sentence. It receives text as input.
Fill Mask: useful when you want to fill the sentence at the masked position. It receives text as input.
Question Answering: useful when you need to answer a question based on a given context.
Image Classification: useful when you want to know the class of the image. It receives image_path as input.
Object Detection: useful when you want to detect the objects in a photo. It receives image_path as input.
Colorization: useful when you want to colorize a photo. It receives image_path as input.
Image Super-Resolution: useful when you want to create a high-resolution image from a low-resolution image.
Image Denoising: useful when you want to denoise a noisy image. It receives image as input.
Image Deblurring: useful when you want to deblur a blurry image. It receives image as input.
Visual Question Answering: useful when you need to answer a question based on a given image.
Image Captioning: useful when you want to know what is inside the photo. It receives image as input.
Text-to-Image Generation: useful when you want to generate an image based on a given description.

Figure A.6: Prompts used for experiments in Tab. 3 and Tab. 4.

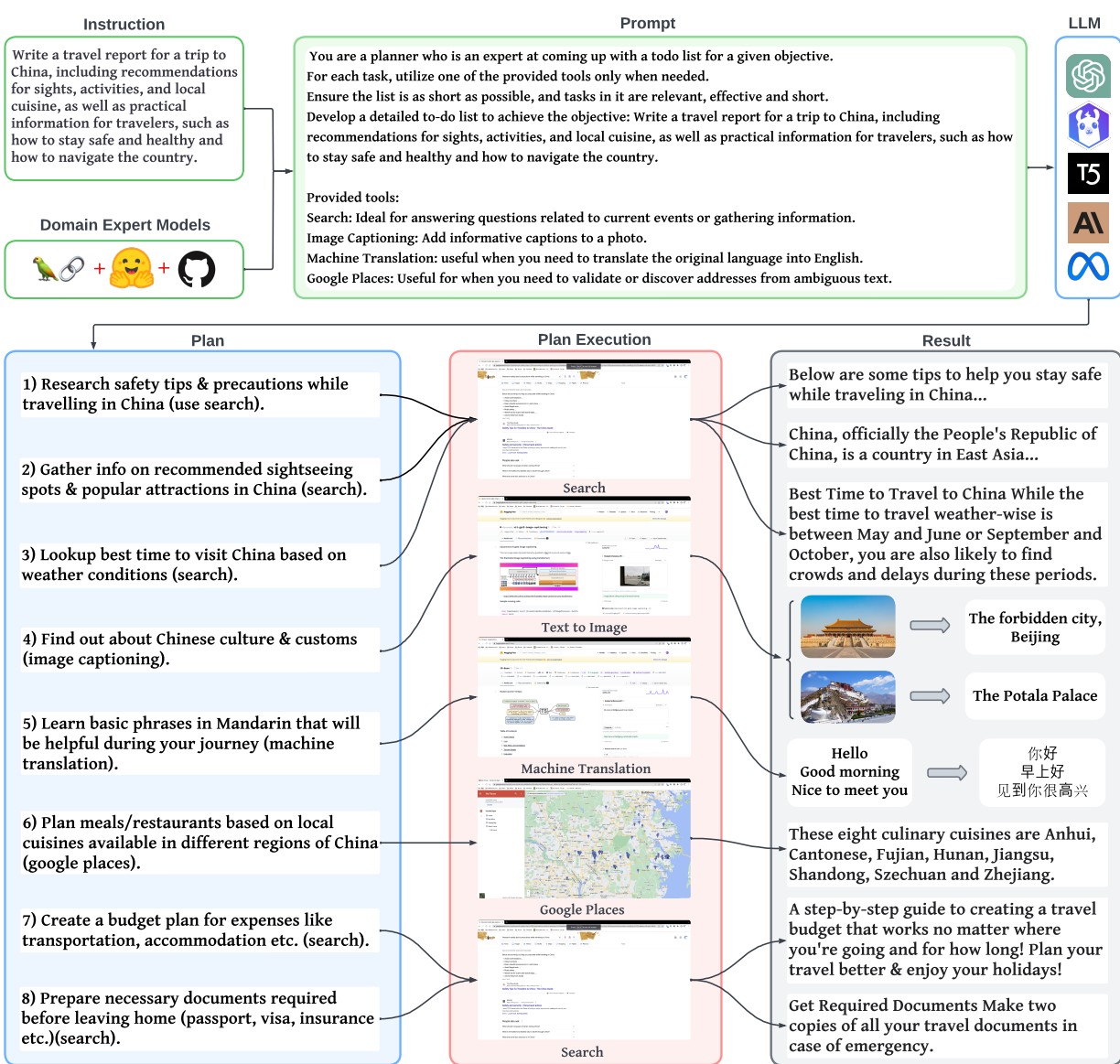

Figure A.7: Another example of open-ended task. OpenAGI is instructed to generate a travel report. The backbone LLM used in this example is Vicuna-7B.

