# OpenReview forum: "OpenAGI: When LLM Meets Domain Experts"
_NeurIPS.cc/2023/Track/Datasets_and_Benchmarks — NeurIPS 2023 Datasets and Benchmarks Poster_

### Official Review · Reviewer_cNMX · 2023-06-21

**Rating:** 6
**Confidence:** 4
**Clarity:** Yes.

**Strengths:**

Open-domain model synthesis is an exciting research area that definitely lacks benchmarks, and so this benchmark platform is very relevant and necessary. The paper is easy to read and mostly clear, especially in the graphical representation of the platform. There are quite a large range of tasks, and the method of creating tasks from data augmentation techniques and combining existing expert models is interesting and could be useful in other settings as well.

**Additional Feedback:**

* Citations are incorred in the first paragraph of page 4 (toolformer is cited with hugginggpt and vice versa)
* It's not known how many parameters GPT 3.5 has, so you can't say it's >175 billion.
* "Possible explanation for this may be Vicuna-7b’s greater data requirements due to its larger model size (as we only used 18 human-labelled data samples for training)." Generally work has shown that larger models need less data, so this is unlikely to be the correct explanation.
* In figure 3 you have users who performan human evaluation, but you don't do any human evaluations in the paper, so this is misleading.
* It's unclear from the paper whether the model is provided with the list of all domain expert models in the prompt regardless of which task it's being asked to complete, or whether it only gets the list of domain expert models that are relevant to the task in the prompt. I think testing with the former settings is more realistic, so if this isn't what's done then the paper is less useful, as it implicitly assumes we know which domain-expert models are relevant for a given task. Which is used?
* More qualitative analysis of model outputs in both successful and failing cases would be useful.
* Including a full list of the tasks and corresponding datasets in the supplementary material would be useful, including which tasks are used for training and which for testing.

**Correctness:**

All the claims in the paper are correct. There's no analysis or representation of the training and testing sets, so it's unclear how similar they are and to what extend models are memorising patterns specific to the platform rather than learning general-purpose behaviour.

**Documentation:**

The code, data and models are released, which is good, but the documentation for the code is very limited, and doesn't show how to benchmark a new model or adjust the training. The code doesn't include a maintenance plan

**Ethics:**

-

**Limitations:**

The authors spend a short paragraph addressing the limitations in the supplementary material, but I think they can spend more effort discussing the limitations of the benchmark in terms of the kinds of tasks it considers, the lack of human evaluations for open-ended tasks, the lack of investigation into how RLTF performs well and what the outputs of T5-flan work, and whether the training method would perform well under distribution shift.

**Opportunities For Improvement:**

I think the paper's weaknesses are in 4 main areas:

# Unnatural task set

While the idea of creating a set of OMS tasks from existing expert models is interesting, these tasks don't seem like natural tasks that users are likely to ask general-purpose AI models. Particularly tasks like "Given two reviews of a movie in English, translate them into German and provide a summary" seem unlikely to be something users would require models to do. The fact that the set of tasks isn't sourced from a study of what users would actually want from OMS models limits the benchmarks usefulness in making sure methods for OMS are actually improving the things we want them to be improving.

This could be improved by creating a more natural task set, particularly where it's immediately obvious which expert models should be used for the task, to test for more open-ended problem solving and planning in LLMs.

# Limited benchmarking of existing models

While 3 models are benchmarked, it would be beneficial to benchmark additional models, both closed-source (claude, gpt4) and open-source (base llama models, larger llama variants, etc.). Additionally, an expert human baseline would be useful to understand how well qualitatively speaking these models are actually doing - the reported scores don't lend themselves to easy interpretation.

# No analysis of successes and failures of methods and models

While it's interesting that T5-Flan with supervised fine-tuning and RLTF outperforms few-shot ChatGPT, there is no analysis of why this is the case, what the plans look like for these models, in what ways they are succeeding or failing, and what open problems remain that these models can't currently do well on. This would enable more targeted future work on this benchmark and help us understand why different methods succeed or fail.

Human evaluations would additionally be useful here to understand how well each model is performing.

# Lack of detail for the benchmark

There is a lack of clarity about how some parts of the benchmark are implemented, for example whether the prompt only includes relevant expert models or all of them, and exactly how the outputted plan of the model is parsed and passed to the expert models. This could be served by better documentation in the github repository.

# Summary

Overall, I think the paper isn't currently worthy of acceptance, but that it could be improved in the rebuttal period by addressing some of the points above. I'd raise my score to 6 if more models were benchmarked and more analysis of the model's performance was done, and I'd raise it further if a more natural distribution of tasks was used and human evaluations were performed.

UPDATE: based on the benchmarking of additional models and analysis in the author's response, I have raised my score to a 6.

**Relation To Prior Work:**

Yes.

**Summary And Contributions:**

This paper introduces the OpenAGI platform for benchmarking LLMs for open-domain model synthesis (OMS): creating plans to utilise multiple expert models to solve complex tasks. The platform has a variety of tasks enabled by a variety of expert models (other AI models, as well as other functionality), where each task consists of a task description and a set of inputs. The LLM takes the task description as input, and produces a plan using the expert models which can be applied to each input in term to produce an output. The platform consists of benchmark tasks with ground-truth outputs which can be used to automatically (using neural evaluation metrics), and open-ended tasks which require human evaluation.

The authors also introduce RLTF, which involves fine-tuning a model with the automated task feedback provided by the ground truth outputs using reinforcement learning.

The paper benchmarks chatGPT, Vicuna-7B and T5-Flan in a zero-shot, few-shot, supervised fine-tuning and RLTF setup. They show that ChatGPT performs well zero-shot and few-shot, while other models generally require some form of additional fine-tuning to perform well. The T5-flan model after supervised fine-tuning and RLTF outperforms chatgpt few-shot despite being much smaller.

---

> ### Author Response · Authors · 2023-08-17
> **Response to reviewer cNMX**
>
> Thank you for the insightful suggestions.
>
> We agree that some of the tasks may seem complex and thus unnatural for human-beings. However, we believe that including these complex tasks in the benchmark is important to test the ability of LLM in terms of complex task-solving. The reason is because based on our experiments with ChatGPT, if it's immediately obvious which expert models should be used for the task, then the LLM can easily predict the expert model and get near perfect (close to 100%) accuracy, which is not very useful in terms of understanding LLM’s ability in solving complex tasks..As a result, our plan is to augment existing benchmarks to develop more complex benchmark tasks.
>
> However, one thing we need to note is that our platform does support more “natural” tasks through the open-ended task solving option, based on which users can freely create any task for the platform to solve. Once the platform solves the user created task and presents the solution to users, users are allowed to provide feedback on the task-solving process by rating the quality of the solution. Once these tasks are created and user feedback is collected, they will be saved into the task database under user consent, which allows the platform to collect and share more and more real-world tasks with their ground-truth labels. We envision this as a dynamic process to establish a virtuous circle between users and AI.
>
> We have tested GPT-4, Claude, and Llama-2-13b.
> Here are the results,
> |            | Claude-2 (zero)  | Claude-2 (few)  | GPT-4 (zero)    | GPT-4 (few)   | Llama-2-13B (zero) | Llama-2-13B (few) | Llama-2-13B (finetune) | Llama-2-13B (RLTF) |
> |------------|-----------|-----------|-----------|-----------|-------------|-------------|-------------|-------------|
> |Clip    | 0.0       | 0.304     | 0.306     | 0.306     | 0.0         | 0.0         | 0.0         | 0.303       |
> | Bert   | 0.229     | 0.321     | 0.379     | 0.631     | 0.255       | 0.150       | 0.572       | 0.350       |
> | ViT    | 0.686     | 0.701     | 0.727     | 0.648     | 0.251       | 0.748       | 0.702       | 0.772       |
> | Overall   | 0.305     | 0.442     | 0.471     | 0.528     | 0.169       | 0.299       | 0.425       | 0.475       |
>
>
> Based on our evaluations, finetuning plays a significant role in enhancing the output quality of smaller, open-source LLMs. Notably, in zero- and few-shot scenarios, these LLMs often struggle to grasp the given instructions, leading to subpar outputs. However, after fine-tuning, there is a marked improvement in quality. Furthermore, our findings suggest that compared to finetuning, RLTF is more effective for text-to-image generation tasks. This can be attributed to the exploratory nature of Reinforcement Learning (RL), which increases the likelihood of producing models optimized for such tasks. Moreover, RLTF is initialized from the fine-tuned weights, which also advances its performances.
>
> In terms of benchmarking complex task-solving, human evaluation falls short in several ways. For example, the augmented images may only pose unnoticeable differences from the original image, which presents challenges for manual comparison, making algorithmic comparison more reliable. Furthermore, the need to interpret multilingual results requires translators, complicating the process of manual labeling since it is challenging to find qualified multilingual annotators.
>
> We confirm that the prompt includes all expert models.
>
> In our approach, we utilized a parser built on GPT-3.5 . The specific prompt we employed reads as follows: “You are a key phrase extractor who is able to extract potential module names from the given context. You have already known all the module names in the full module list. The full module list is: [Image Classification, Colorization, Object Detection, Image Deblurring, Image Denoising, Image Super Resolution, Image Captioning, Text to Image Generation, Visual Question Answering, Sentiment Analysis, Question Answering, Text Summarization, Machine Translation]. Given the following context: '{}'. Please extract a module sequence from this context and remove module names which do not exist in the full module list from this sequence. Output the module sequence after filtering as the format of 'module: module1, module: module2, module: module3, etc...'.” Comprehensive details about this process can be found in the supplementary material and our Github repository.
>
> We have added the discussion of the limitations of the benchmark tasks and the dataset as well as open-ended tasks into the limitation section.

---

> > ### Author Response · Authors · 2023-08-17
> > **contd**
> >
> > Benchmarking a new LLM is streamlined with our proposed dataset, given that all other components are meticulously designed within OpenAGI. We have included a new notebook in our repository to assist researchers in evaluating their own LLMs.
> >
> > We are planning to implement a maintenance plan for our project to ensure its ongoing relevance, efficiency, and user trust. Key components of this plan include scheduled updates, a dedicated feedback channel, regular code reviews, updated training materials, performance monitoring, and community engagement initiatives. Additionally, we will address any critical bugs or security concerns swiftly, collaborate with like-minded projects, and foster a diverse and inclusive environment. Our goal is to proactively manage the project, maintaining its quality and serving our community effectively.
> >
> > We have fixed the citation issue as well as the description about GPT 3.5 in our paper.
> >
> > We believe that Flan-T5 surpasses Vicuna-7B in performance because it was finetuned on all parameters. In contrast, due to GPU memory constraints with Vicuna-7B, we only employed the LoRA finetuning method. To validate our hypothesis, we applied LoRA fine-tuning to Flan-T5. Here are the results,
> > | Metric  | Flan-T5 (LoRA)  |
> > |---------|----------------|
> > | Clip    | 0.0             |
> > | Bert    | 0.118         |
> > | Vit     | 0.647          |
> > | Overall | 0.255       |
> > We have updated the analysis with the latest Flan-T5 results for a more comprehensive comparison.
> >
> > Regarding the domain expert models in the prompt, we confirm it is the former setting, i.e., the model is provided with the list of all domain expert models in the prompt regardless of which task it's being asked to complete.
> >
> >
> > Regarding the pipeline figure in the paper, to clarify, the arrow labeled 'benchmark evaluation' is intended for assessing the benchmark dataset, while the 'human evaluation' arrow pertains to the evaluation of open-ended tasks. Besides, we used dotted-line instead of solid-line for human evaluation in the figure to distinguish the two for clarity. We will provide further clarification on this point in the relevant section. We also include a full list of the tasks and corresponding data samples in supplementary, and indicate which tasks are used for training and which for testing.

---

> > ### Comment · Reviewer_cNMX · 2023-08-18
> >
> > Thank you for the additional benchmarking of new models and the clarifications you gave. Given the new analysis and models, I am increasing my score to a 6.

---

> > > ### Author Response · Authors · 2023-08-27
> > >
> > > Dear reviewer, we sincerely appreciate your support. Thank you.

---

### Official Review · Reviewer_9AkB · 2023-07-12
**The authors open-source OpenAGI benchmark. It is a supplement to complex, multi-step tasks.**

**Rating:** 8
**Confidence:** 4
**Correctness:** Yes
**Clarity:** Yes

**Strengths:**

The authors open-source the benchmark, includes code, data, evaluation methods and demo.

**Additional Feedback:**

1）The difficulty of nonlinear task is not clear. In my point of view, the examples listed in the paper are easier than some linear tasks. The ability to plan a processable pipeline should be a big challenge.
2）What is the fine-tune technique used in the paper? Make a more clear explanation between fine-tune and RLTF.
3)  More LLMs should be added in experiments.

**Documentation:**

Yes

**Limitations:**

The authors have addressed the limitations.

**Opportunities For Improvement:**

It is better the authors can give more clear categories of the tasks with different dimensions, for example, based on the difficulty level.

**Relation To Prior Work:**

Yes

**Summary And Contributions:**

The authors introduce  a benchmark called OpenAGI with complex, multi-step tasks. A Reinforcement Learning from Task Feedback (RLTF) mechanism is proposed to improve the LLM's ability. The experiments with three well-known LLMs show that the smaller-scale LLMs with RLTF can potentially outperform larger models.

---

> ### Author Response · Authors · 2023-08-16
> **Response to reviewer 9AkB**
>
> Thank you for the insightful suggestions.
>
> It is really useful to consider the task's difficulty level. We will introduce a difficulty level based on the number of models invoked in the human-labeled solutions.
>
> While nonlinear tasks may appear simpler, as demonstrated in Section 5.6, only GPT is capable of tackling these tasks. This underscores the challenges they present in reasoning and planning for LLMs.
>
> Apologies for any confusion in terms of fine-tuning. In the fine-tuning process, we use 18 human-labeled data points as the ground-truth labels to refine a given LLM. Conversely, for RLTF, we employ the same tasks but without the human labels. Since RLTF learns through a trial-and-error paradigm, the LLM-generated solutions are executed, and their performance are used as rewards to subsequently guide the adjustment of the LLM weights.
>
> Furthermore, we have added GPT-4, Claude, and Llama-2-13b.
> Here are the results,
> |            | Claude-2 (zero)  | Claude-2 (few)  | GPT-4 (zero)    | GPT-4 (few)   | Llama-2-13B (zero) | Llama-2-13B (few) | Llama-2-13B (finetune) | Llama-2-13B (RLTF) |
> |------------|-----------|-----------|-----------|-----------|-------------|-------------|-------------|-------------|
> |Clip    | 0.0       | 0.304     | 0.306     | 0.306     | 0.0         | 0.0         | 0.0         | 0.303       |
> | Bert   | 0.229     | 0.321     | 0.379     | 0.631     | 0.255       | 0.150       | 0.572       | 0.350       |
> | ViT    | 0.686     | 0.701     | 0.727     | 0.648     | 0.251       | 0.748       | 0.702       | 0.772       |
> | Overall   | 0.305     | 0.442     | 0.471     | 0.528     | 0.169       | 0.299       | 0.425       | 0.475       |

---

### Official Review · Reviewer_fmdC · 2023-07-21
**Interesting platform but I think more work/research needs to be done to make it a really useful benchmark/dataset**

**Rating:** 6
**Confidence:** 3
**Correctness:** The dataset was constructed in a soun…

**Strengths:**

- RLTF is original, novel, and works well
    - Although, it is very similar to RLHF, so I believe that RLHF should be discussed. It is still a novel contribution though.
- Useful platform/tool for developers
-Benchmarked existing LLMs

**Additional Feedback:**

- Ultimately, I don't see a significant benefit of benchmarking LLMs in this way as I don't believe it is a route towards AGI nor do I think it will be something that needs to be benchmarked. This makes the contribution/potential impact seem limited.
- Some of the claims that LLMs seem like a path towards AGI when augmented with expert models are a bit too strong/unstantiated
- I believe that in its current state this is more of a research tool/software project turned into a benchmark than an actual standard benchmark.
- It is possible that this work could have a very high impact and that it is the first benchmark of its kind, I just find this scenario unlikely.

**Clarity:**

The paper has a lot of grammatical issues and the flow is off in several places. I also believe a lot of things were not included in the paper (they were in the supplementary) that should have been in favor of other things (i.e. limitations, RLTF equations)

**Documentation:**

There is not sufficient detail on these dataset details. There is a section called "Dataset Documentation and Data Samples for Benchmark Tasks", but it does not contain a lot of this information.

**Limitations:**

I think the authors could do a better job at addressing potential limitations and negative impacts. For example, I think more comprehensive testing could be done as to the negative things that can be done with the current framework. I also think the detrimental effects mentioned in the limitations should be significantly expanded upon and have research to back it up, especially if this technique is a path towards AGI.

**Opportunities For Improvement:**

- Still ultimately has the issue of a limited amount of usable expert models
- Used CLIP/Bert/ViT score as ground truth - this also adds noise to the evaluation process
- Writing style could be improved upon
- The authors could potentially introduce more novel architectures or approaches

**Relation To Prior Work:**

Yes, although I'm not sure why AutoGPT isn't mentioned.

**Summary And Contributions:**

Authors propose research platform including benchmarks and models called OpenAGI. Also include novel RLTF technique.

---

> ### Author Response · Authors · 2023-08-16
> **Response to reviewer fmdC**
>
> Thank you for your valuable feedback and suggestions. We have included additional related work on RLHF in our 'Related Work' section, since it is closely related to our proposed method, RLTF. We’ve also added AutoGPT, BabyAGI and many other related Github repos into this section.
>
> We acknowledge the importance of broadening the diversity and range of expert models to enhance the adaptability and scalability of our platform. However, we have limited the size of such a model set for a few reasons. Firstly, maintaining a manageable set size allows for feasible benchmarking, given that the selection of expert models is largely dictated by the available datasets. For instance, while we could incorporate a search engine into the model set, there is not a suitable public dataset for benchmarking such an expert model. Secondly, the Large Language Models (LLMs) like Flan-T5 and Llama have constraints on context length, which are limited to 512 tokens and 2k tokens, respectively. Adding more models could easily exceed these constraints, especially when model names or descriptions are included in the prompt.
>
> Considering the new architecture issue, on one hand, this work primarily emphasizes the data and benchmark perspective. Consequently, the goal is not to design new LLM architectures. Instead, we evaluated various existing and widely used LLMs under different learning schemas. Our findings suggest that there is substantial potential for research in this area, given that many existing methods underperform. On the other hand, except for zero-shot learning, few-shot learning, and fine-tuning, we have implemented an innovative RLTF mechanism for modeling training to show the performance of various LLMs under different learning schemas. Furthermore, we have benchmarked even more LLMs for greater comprehensiveness, please kindly refer to the next part of this response for detailed results.
>
> Given the limitations highlighted in our study, we believe it is essential for the community to establish a benchmark for LLMs on complex task solving, which is the very purpose of this research. Subsequent research on these limitations can then be built upon the dataset, benchmark, and framework. Moreover, we have also provided the constrained generation framework in the project to help mitigate such negative effects, which allows users to freely configure the constraints in the plan generation process of LLMs so as to regulate the generated plans.
>
> We also appreciate your skepticism about the value of benchmarking LLMs and domain expert models on complex task solving as a route towards AGI, and we believe that healthy debates on this topic is important. We acknowledge that as an initial effort, there is certainly room to improve our project’s quality and adoption as a widely used benchmark. However, OpenAGI is not a tool or software project, instead, we focus on evaluating the reasoning and planning ability of LLMs. Moreover, we respectfully disagree that this offers limited contribution or impact. This benchmarking serves several important purposes:\
> •	It enables standardized measurement of capabilities and progress across the research community. Lack of unified benchmarks hampers advancement.\
> •	Analyzing model performance on composed tasks provides insights into emergent abilities from combining skills - an essential aspect of general intelligence.\
> •	Benchmarking on expansive tasks helps identify current method limitations to guide research directions.\
> •	Availability of an open benchmark facilitates faster reproduction and apples-to-apples comparison of techniques by different groups.\
> •	Quantifying capabilities on complex human-relevant tasks, even if imperfect, keeps research grounded in real-world utility.

---

> > ### Author Response · Authors · 2023-08-16
> > **Contd**
> >
> > While no single benchmark will encapsulate all facets of AGI, we believe our work provides a useful framework for the community by allowing principled measurement and investigation of key aspects of intelligent behavior. More importantly, our project takes a very inclusive design from the beginning, which allows future AI advancement to be easily integrated into the framework. More specifically, we understand that there may be different AI development philosophies in the community, such as connectionism, symbolism, empiricism and rationalism, however, no matter for neural models, symbolic models, planning models, reasoning models or any other models, they can be easily integrated into the framework as expert models, future improvements on LLM such as its reasoning and planning ability can also be integrated by upgrading the LLM backbone. Furthermore, the framework also supports human interaction and intervention in the process. As a result, our project does not exclude any possible approach towards AGI.
> >
> > However, we agree that further advances beyond benchmarking will be needed to achieve AGI. To that end, we are open-sourcing OpenAGI to engage the community. We believe that our platform can accelerate research in this field as an initial proxy for human intelligence. Besides, we are always open to feedback from the community on how to maximize its usefulness.
> >
> > Besides, as mentioned in the above part of the response, we appreciate your suggestion to consider other LLMs, so we have added GPT-4, Claude, and Llama-2-13b, and here are the results,
> > |            | Claude-2 (zero)  | Claude-2 (few)  | GPT-4 (zero)    | GPT-4 (few)   | Llama-2-13B (zero) | Llama-2-13B (few) | Llama-2-13B (finetune) | Llama-2-13B (RLTF) |
> > |------------|-----------|-----------|-----------|-----------|-------------|-------------|-------------|-------------|
> > |Clip    | 0.0       | 0.304     | 0.306     | 0.306     | 0.0         | 0.0         | 0.0         | 0.303       |
> > | Bert   | 0.229     | 0.321     | 0.379     | 0.631     | 0.255       | 0.150       | 0.572       | 0.350       |
> > | ViT    | 0.686     | 0.701     | 0.727     | 0.648     | 0.251       | 0.748       | 0.702       | 0.772       |
> > | Overall   | 0.305     | 0.442     | 0.471     | 0.528     | 0.169       | 0.299       | 0.425       | 0.475       |
> >
> > Furthermore, we will provide a comprehensive list of tasks along with their corresponding data samples in the "Dataset Documentation and Data Samples for Benchmark Tasks" section of the supplementary material. We will also specify which tasks were used for training and which were used for testing to offer readers a clearer understanding of our dataset.

---

> > > ### Comment · Reviewer_fmdC · 2023-08-27
> > >
> > > I really appreciate the effort in addressing my and the other reviewers' concerns as well as any additional experimentation. After reviewing additions to the paper and reasoning, I am confident in increasing my score to a 6 for now. I think the most difficult challenge with the novelty of this work is the plethora of people doing something similar (although I realize many of them began after the submission deadline).
> > >
> > > Unfortunately, I still don't see a discussion on RLHF in the related works. Is it possible I am missing this discussion or was it not added yet? Similarly, I could not find the table shown above in the paper.

---

> > > > ### Author Response · Authors · 2023-08-27
> > > >
> > > > Dear reviewer, thank you very much for your support. We have added the discussion of RLHF to the related work. We also added the result tables of GPT-4, Claude-2, and Llama-2 into the paper. The PDF file has been updated. Thank you.

---

### Official Review · Reviewer_Zkzc · 2023-07-21
**Review for Submission 305**

**Rating:** 7
**Confidence:** 3
**Correctness:** NA
**Clarity:** NA

**Strengths:**

1. The paper proposes an interesting open-source platform, OpenAGI, to evaluate the LLMs' ability to do planning and problem-solving for multi-step tasks.
2. The quantitative benchmarks proposed offers a good way to evaluate the LLMs.
3. The RLTF method is very interesting and effective based on the evaluation result.

**Additional Feedback:**

The paper uses Vicuna model but does not correctly cite it: https://lmsys.org/blog/2023-03-30-vicuna/.

**Documentation:**

NA

**Limitations:**

Please refer to "Opportunities For Improvement"


**Opportunities For Improvement:**

1. In the benchmark tasks, it seems the image-text pair in, image-out combination is missing, which seems to be an important use case.
2. The 18 human-labeled data used in the RLTF experiments seem a bit small. It would be nice to see if more human-labeled data can improve the performance of open-source models to uncover the potential of open-source models.
3. Considering the good performance of the Vicuna-7b and Flan t5 used in the paper, it would be nice to see if a larger open-source model can outperform the ChatGPT


**Relation To Prior Work:**

NA

**Summary And Contributions:**

The paper proposes a platform, OpenAGI, that helps evaluating the planning and task-solving ability of LLMs to tackle the multi-step, real-world tasks. It proposes two set of tasks, benchmark tasks and open-end tasks to evaluate the LLMs. It also proposes a RLTF approach to further improve the performance of the LLMs on the task-solving ability. Finally, it evaluates three LLMs and demonstrates the effectiveness of the proposed method.

---

> ### Author Response · Authors · 2023-08-16
> **Response to reviewer Zkzc**
>
> Thank you for your great suggestions.
>
> Indeed, the tasks we selected originate from existing CV or NLP tasks, to which we added image or text augmentations. Given the current lack of datasets for image-text input with image output tasks, we haven't incorporated such tasks. However, we will continue monitoring developments in this area and we will aim to expand our tasks dataset accordingly when such datasets become available.
>
> We utilized only 18 tasks because the total available number of tasks is only 180 (we used 10% for training), and the augmented tasks share a lot of similarities. These 18 tasks encompass all the domain expert models in our set. Actually, we consider it as an important advantage of our RLTF method that just needs a small number of tasks for training, which shows that it is sufficient for the LLM to learn and make high-quality inferences.
>
> Besides, we have added Llama-2-13B, as well as Claude-2.0, and GPT-4 into the experiments. Here are the results,
> |            | Claude-2 (zero)  | Claude-2 (few)  | GPT-4 (zero)    | GPT-4 (few)   | Llama-2-13B (zero) | Llama-2-13B (few) | Llama-2-13B (finetune) | Llama-2-13B (RLTF) |
> |------------|-----------|-----------|-----------|-----------|-------------|-------------|-------------|-------------|
> |Clip    | 0.0       | 0.304     | 0.306     | 0.306     | 0.0         | 0.0         | 0.0         | 0.303       |
> | Bert   | 0.229     | 0.321     | 0.379     | 0.631     | 0.255       | 0.150       | 0.572       | 0.350       |
> | ViT    | 0.686     | 0.701     | 0.727     | 0.648     | 0.251       | 0.748       | 0.702       | 0.772       |
> | Overall   | 0.305     | 0.442     | 0.471     | 0.528     | 0.169       | 0.299       | 0.425       | 0.475       |
>
> We can see that the performance of Llama-2-13B on zero- and few- shot appears to be nearly close to that of GPT-3.5, though GPT-4 is still a fairly strong model among others.

---

### Official Review · Reviewer_6ZWy · 2023-08-02

**Rating:** 5
**Confidence:** 3
**Correctness:** The pipeline seems constructed in a s…

**Strengths:**

- The topic of LLM-based agents is important and worth-exploring

**Additional Feedback:**

- The authors did not sign the license, nor include the license of the code that it is based on (e.g. Restormer)
- The authors should consider including GPT-4 as one of the backbone LLMs. Given the increasingly open availability of GPT-4 access, there isn't a reason why GPT-4 shouldn't be included.

**Questions regarding the experiments:**
- I am fairly confused when I am reading through the experiment section. It remains unclear what tasks the authors sought to benchmark. Given the presence of numerous vision-language alignment metrics, such as CLIP, BERT, and ViT scores, one might infer that the tasks were related to text-to-image alignment. If that is the case, further details (e.g. size of the evaluation dataset) on the experiment's execution are needed to fully understand the methodology employed.
- As I have mentioned above, CLIP, BERT, and ViT scores are metrics mainly used in image-captioning - to test text-to-image alignments. They are not perfect nor comprehensive enough for evaluating the generalization of your method in any way. Consider incorporating a wider array of metrics and conducting evaluations on a broader range of downstream tasks. This will enhance the comprehensiveness and robustness of your evaluation, thereby strengthening the paper's validity and reliability.
- `` Given that only 18 data samples were utilized, it suggests that RLTF is an efficient and viable method for task alignment. `` Did you report the training performance in Table 1? If not, what is the size of the evaluation set?
- Section 5.5 suggests a huge variance coming from changing the prompts being used. You should similarly perform this kind of ablation study on the RLTF method as well.

**Clarity:**

The paper's readability could be enhanced significantly, as it currently lacks certain essential details, making it difficult to follow. Improvements in the presentation and organization of the content are necessary to bolster the paper's quality. The details regarding the RLTF algorithm and the experiment setup are not included in the main text, and they shouldn't be put in the appendix.

**Documentation:**

Looks like there are a sufficient amount of details provided on the GitHub repo to reproduce the results. I have yet to confirm if the code can work without any issues.

**Limitations:**

The discussion of limitations should be moved to the main text.

**Opportunities For Improvement:**

- **The paper's clarity could be improved significantly**. Please refer to the 'additional feedback' section for a list of queries regarding your experiments. The scope of the experiments and evaluations conducted in this study seems somewhat limited in my view.
- The project does not give the impression of being long-term. Questions arise regarding the software quality, maintainability, and consequently, the long-term reliability of this platform. There seems to be a lack of rigorous QA/QC, evidenced by the absence of GitHub actions, build, and test routines. No active updates have been noted in the GitHub Repo since mid-June, hinting at a somewhat static user community
- In 'Plan Execution' as presented in Figures 1 and 2, the authors offer screenshots from GitHub Repos (not their own) in an attempt to demonstrate the pipeline's functionality. This method lacks convincing power. A more compelling approach could be to replace screenshots with informative data visualizations.

**Relation To Prior Work:**

There are many existing works on LLM-based agents (i.e. where LLM is used in planning and completing downstream tasks), I think the authors should at least discuss the existences of publicly-available GitHub repos such as Auto-GPT and "ChatGPT for Robotics" by Microsoft at the related work section. I don't see how this work is too different from those LLM-based agents. I think you could similarly call the different configurations (e.g. image generation, web search) in Auto-GPT a set of "domain expert skills".

**Summary And Contributions:**

This work introduces OpenAGI, an open-source AGI research platform designed for multi-step, real-world tasks. The authors have argued that the existing works have three major problems: (1) only work for a fixed number of models (2) limited to solving tasks with linear task-planning solutions, meaning that each sub-task must be completed before the next sub-task can start, no modalities, etc. (3) only provide qualitative results. The benchmark tasks (as described in Figure 1) are those that have specific objectives and their performance can be gauged by a specific benchmark, such as image-to-text or text-to-image. The open-ended tasks (as described in Figure 2), as the name suggested, are those more oriented towards subjective evaluations, such as music and poem generation. This paper also discuss the model fine-tuning method of Reinforcement Learning from Task Feedback (RLTF), and some experiment results have been provided.

---

> ### Author Response · Authors · 2023-08-16
> **Response to reviewer 6ZWy**
>
> Thank you for raising these important points.
>
> We’ve added the dataset license CC BY-NC-SA 4.0 into the github repo and included the license of Restormer in it as well.
>
> Before submitting, none of the authors had access to the GPT-4 API key, but now it has open access to all users, so we have included GPT-4 in our main experiments. Moreover, we also included Claude-2.0 and Llama-2-13B to further explore their performances.
> Here are the results:
> |            | Claude-2 (zero)  | Claude-2 (few)  | GPT-4 (zero)    | GPT-4 (few)   | Llama-2-13B (zero) | Llama-2-13B (few) | Llama-2-13B (finetune) | Llama-2-13B (RLTF) |
> |------------|-----------|-----------|-----------|-----------|-------------|-------------|-------------|-------------|
> |Clip    | 0.0       | 0.304     | 0.306     | 0.306     | 0.0         | 0.0         | 0.0         | 0.303       |
> | Bert   | 0.229     | 0.321     | 0.379     | 0.631     | 0.255       | 0.150       | 0.572       | 0.350       |
> | ViT    | 0.686     | 0.701     | 0.727     | 0.648     | 0.251       | 0.748       | 0.702       | 0.772       |
> | Overall   | 0.305     | 0.442     | 0.471     | 0.528     | 0.169       | 0.299       | 0.425       | 0.475       |
>
> After submitting our paper to NeurIPS, many of the authors pursued summer internships at industry companies, leading to infrequent updates to the repository over the summer. Besides, we are awaiting feedback from NeurIPS reviewers. Based on the reviews and our new progress, we have updated the repo with new implementations and experiments. We will continue maintaining and updating the repo in the years to come, such as adding an AI constitution to the project as we mentioned in the paper.
>
> We also provide more clarification in describing our experimental methodology: In this work, we try to benchmark the tool planning ability of LLM by constructing complex vision-language tasks as the dataset for evaluation. Specifically, we use a lot of image-, and text- augmentation methods to augment the original data, then require the LLM to select the useful tools in a reasonable order to restore the original data. Therefore, our goal was to benchmark a diverse set of tasks spanning multiple modalities, not just text-to-image alignment.
>
> To clarify, we constructed 185 unique multi-step tasks in OpenAGI across 6 categories:\
> •	Image in, image out\
> •	Image in, text out\
> •	Text in, image out\
> •	Text in, text out\
> •	Image-text pair in, text out\
> •	Text-text pair in, text out\
> These categories cover a range of data types and required skills. We used 10% of the tasks (about 18) for training, and we sample another 10% from the remaining tasks for testing. We repeat the sampling and testing process for several rounds to calculate the average performance.
> The choice of evaluation metric depends on the task output type:\
> •	CLIP score was used only for text-to-image generation tasks\
> •	BERT score was used for tasks with text output\
> •	ViT score was used for tasks with image output\
> In hindsight, we have provided more examples of the diverse task types and specifics on the composition of the evaluation set. The tasks were not solely focusing on text-to-image alignment. We sincerely appreciate it that you highlighting this point - it helps us to improve the clarity and thoroughness of our methodology description.
>
> The performance metrics for RLTF are derived from the test set, we sample 10% of the remaining task (14 tasks) for testing. We repeat the sampling and testing process for several rounds to calculate the average performance. These tasks were randomly chosen, distinct from the training tasks.
> The improved performance could be attributed to:
> 1) Initializing RLTF with weights from the fine-tuned model, which were already of high quality;
> 2) RL introduces the ability of exploration when generating models, which is a crucial factor;
> 3) The training tasks encompass all expert models in our dataset, thereby better assisting the learning of LLM.
>
> Furthermore, regarding the figures in the paper, we have replaced those screenshots with data visualizations from our own experiments. Please check the updated content in our GitHub repository. We will also update the figures to the paper. Thank you.

---

> > ### Comment · Reviewer_6ZWy · 2023-08-19
> > **Follow-ups**
> >
> > Thank you to the authors for their clarifications. In this context, I believe that the composition of evaluation sets lacks task diversity and quantity. I question whether scheduling an image processing workflow can truly assess the capability of a tool-use LLM agent. Wouldn't it be an overstatement to call it 'OpenAGI'? And I think you should do a more thorough study of the related work, as I have originally suggested. There are so many LLM agent studies available nowadays.
> >
> > The added evaluations using GPT-4, Claude-2, and Llama-2 are very much appreciated.

---

> > > ### Author Response · Authors · 2023-08-19
> > >
> > > We would like to thank you for the follow up communications.
> > >
> > > We appreciate your suggestion of adding more related works and we agree LLM agents for complex task-solving has been an important topic recently. Actually, this also demonstrates the importance of our project as an attempt to benchmark the complex task-solving ability of LLMs. However, most of the LLM agent works appeared very recently just in the past one or two months and mostly after the NeurIPS submission was due. During the development period of our project from Fall 2022 up to the NeurIPS submission deadline, we were unaware of most of the current LLM agent works except for those ones that we have already cited in the paper. However, as mentioned, we appreciate your suggestion and we will add more related works on LLM agent studies, which should be a easy fix to the paper.
> > >
> > > We also appreciate your question on 'OpenAGI' and we hope to provide more discussions and clarifications on this. We honestly believe OpenAGI is a legitimate name for the project, and this is because of following three reasons.
> > >
> > > 1) Our project not only supports benchmark tasks but also supports open-ended tasks. The benchmark tasks are important because we hope to contribute to the community with a standard way of evaluating the complex task-solving ability of LLMs. Based on the experimental results shown in the paper and in the discussion period, even 'scheduling an image processing workflow' is not an easy task for most state-of-the-art LLMs such as GPT-4 and LLaMA. Furthermore, the open-ended task mode allows users to create basically any task using natural language for the platform to solve. Once the platform solves the user created tasks and presents the solution to users, users are allowed to provide feedback on the task-solving quality. Once these tasks are created and user feedback is collected, they will be saved into the task database under user consent, which allows the platform to collect and share more and more real-world tasks with their ground-truth labels. As a result, by design, the platform has no restriction on the type of problems to solve, and we envision this as a dynamic process to establish a virtuous circle between humans and AI.
> > >
> > > 2) Our project takes a very inclusive design from the beginning, which allows future AI advancement to be easily integrated into the framework. More specifically, we understand that there may be different AI development philosophies in the community, such as connectionism, symbolism, empiricism and rationalism, however, no matter for neural models, symbolic models, planning models, reasoning models or any other models, they can be easily integrated into the framework as expert models, and future improvements on LLM such as its reasoning and planning ability can also be integrated by upgrading the LLM backbone. Furthermore, the framework also supports human interaction and intervention in the process. As a result, our project does not exclude any possible approach towards AGI.
> > >
> > > 3) We are not the only one to follow an AGI type of naming for relevant projects. Except for our project, there are other recent projects such as BabyAGI, SuperAGI, MiniAGI, BigAGI, etc. which also follow an AGI type of naming. Our project has differences from these ones in that we support both benchmarking tasks and open-ended tasks, as well as an innovative Reinforcement Learning from Task Feedback (RLTF) mechanism to enable LLMs' self-improvement from task-solving experiences. However, as mentioned above, we appreciate your suggestion and we will cite and discuss these projects in the related work of the paper. Of course, we will also cite and discuss other recently released projects that do not have an AGI type of naming in our paper.
> > >
> > > Finally, we would like to thank you for acknowledging our evaluations on GPT-4, Claude-2, and Llama-2.

---

> > > > ### Author Response · Authors · 2023-08-27
> > > >
> > > > Dear reviewer, following your suggestion, we have expanded the discussion of related works and added the suggested citations. We have also added the results of GPT-4, Claude-2, and Llama-2 into the paper. Thank you.

---

> > > > > ### Comment · Reviewer_6ZWy · 2023-08-28
> > > > > **thanks for the rebuttal**
> > > > >
> > > > > Thanks to the authors for their time and effort spent on the rebuttal. Your response to my and other reviewers' questions has addressed some of my questions. The scope of the work appears large in the introduction but small in the actual implementation. The size of the evaluation is small, which could be problematic in gauging the actual ability of the models. Though testing LLM's ability to perform image-editing-related tasks is useful.
> > > > >
> > > > > Upon reconsideration, I will increase my rating to 5.
> > > > >
> > > > > P.S. I'm uncertain whether an additional page is permitted in the rebuttal this year, as it was last year.

---

> > > > > > ### Author Response · Authors · 2023-08-29
> > > > > >
> > > > > > Dear reviewer, we sincerely appreciate your support. We confirm that one additional page is permitted, since the call for paper mentioned "If your submission is accepted, you will be allowed an additional content page for the camera-ready version." We also ensure that the project will be a continuous effort by ourselves and the community engagement, and many exciting things will be developed on top of it. Thank you.

---

> > > > > > > ### Comment · Reviewer_6ZWy · 2023-08-30
> > > > > > > **follow-up**
> > > > > > >
> > > > > > > I saw multiple submissions under review did the same thing, yeah I guess it is not a big deal then. But the sentence you quoted specifies that an additional page is only allowed if your submission is "accepted", not "submitted".

---

> > > > > > > > ### Author Response · Authors · 2023-08-30
> > > > > > > >
> > > > > > > > Dear reviewer, we very much appreciate your clarification. Thank you.

---

### Decision · Program_Chairs · 2023-09-22

**Decision:**

Accept (Poster)

**Comment:**

The paper proposes OpenAGI, a framework for testing and training existing and future AGI models (here, for example, those that can take in or output either text or images).
OpenAGI is based on the idea that large language models (LLMs) now can read task instructions in a sentence (e.g. "restore this image", and generate a list of steps for solving the task (akin to AutoGPT). The framework is a closed loop that starts from (1) reading task instructions; (2) generating a list of steps toward the solution; (3) utilizing existing tools to execute each step; and (4) generating results.
In such a closed loop, LLMs can be fine-tuned using a set of objective measures defined by the generated results, i.e. no humans in the loop or Reinforcement Learning from Task Feedback (RLTF).

This paper is more like ViperGPT and AutoGPT, yet, applied to testing and training AI models themselves.
The work is quite thorough: exploring 6 different input/output settings; 6 different LLMs. The framework also shows that significant performance improvements can be observed for Vicuna-7b and Flan-T5-Large when fine-tuning or using RLTF compared to zero-shot and few-shot learning methods.

The work is solid and thorough. The framework and its code can practically contribute to the community.